# A cholesterol-responsive hepatic tRNA-derived small RNA regulates cholesterol homeostasis and atherosclerosis development

Xiuchun Li[1], Rebecca Hernandez [1], Xudong Zhang [2,3], Sijie Tang [1], Xiaohong Yuan [4], Jing Wu[4], Kathy Pham[1], Hukam C. Rawal [5], Erica C. Heinrich [1], Shenglong Zhang [4,6,7], Qi Chen [2,3], Tong Zhou [5] & Changcheng Zhou [1] ✉

Transfer RNA-derived small RNAs (tsRNAs) have emerged as crucial players in diverse biological processes. Yet, their involvement in lipid metabolism and cardiovascular disease remains elusive. Using an advanced PANDORA-seq method, we identify tsRNA-Glu-CTC as the most abundant tsRNA in mouse liver. Intriguingly, tsRNA-Glu-CTC is cholesterol responsive. Overexpression of tsRNA-Glu-CTC elicits hypercholesterolemia and hepatic steatosis, whereas its knockdown protects against diet-induced hypercholesterolemia and atherosclerosis in mice. Mechanistically, tsRNA-Glu-CTC regulates key hepatic lipogenic genes including *Srebp2*, a master regulator of lipid metabolism. tsRNA-Glu-CTC interacts with SREBP2 to regulate its own transcription through an E-box motif. We further identify site-specific RNA modifications of endogenous tsRNA-Glu-CTC by a mass spectrometry-based MLC-seq and demonstrate the modified tsRNA-Glu-CTC as a more potent regulator of cholesterol homeostasis compared to its unmodified synthetic counterpart. Collectively, our study reveals an important role of a liver-enriched tsRNA in lipid metabolism and cardiovascular health, opening new therapeutic avenues for cardiometabolic disease.

Cholesterol is an essential component of cell membranes and is also a precursor of numerous steroid hormones and bile acids[1–3]. Cholesterol homeostasis is crucial for normal cellular and biological functions, and aberrant cholesterol homeostasis can lead to the development of many diseases including cardiovascular disease (CVD), neurodegenerative disease, and cancers[2,4]. Cholesterol concentration is tightly regulated by a feedback control system including transcriptional and post-transcriptional mechanisms[1,2]. For transcriptional regulation, sterol regulatory element binding proteins (SREBPs) have been established as the master regulator of lipid homeostasis[1–3]. There are three SREBP isoforms (SREBP1a, SREBP1c, and SREBP2), and SREBP2 preferentially regulates key genes involved in cholesterol biosynthesis

[1]Division of Biomedical Sciences, School of Medicine, University of California, Riverside, CA, USA. [2]Molecular Medicine Program, Department of Human Genetics, University of Utah School of Medicine, Salt Lake City, UT, USA. [3]Division of Urology, Department of Surgery, University of Utah School of Medicine, Salt Lake City, UT, USA. [4]Department of Biological and Chemical Sciences, New York Institute of Technology, New York, NY, USA. [5]Department of Physiology and Cell Biology, University of Nevada, Reno School of Medicine, Reno, NV, USA. [6]Department of Chemistry, University at Albany, State University of New York, Albany, NY, USA. [7]The RNA Institute, University at Albany, State University of New York, Albany, NY, USA. ✉e-mail: changcheng.zhou@ucr.edu

and metabolism including the HMG-CoA reductase (HMGCR) and Proprotein convertase subtilisin/kexin type 9 (PCSK9)[1–3,5,6].

In addition to the well-studied traditional regulators, recent studies have identified many small non-coding RNAs (sncRNAs) especially microRNAs (miRNAs) as key regulators of lipid homeostasis[4,7–11]. For example, miRNA-33 has been demonstrated to regulate cholesterol homeostasis by coordinating with SREBP host genes[8–10]. miRNA-148 can also regulate hepatic LDL receptor (LDLR) and ABCA1 expression to modulate lipoprotein levels in vivo[4,12]. A highly expressed hepatic miRNA, miRNA-122[13,14] has been shown to regulate cholesterol and fatty-acid metabolism[15,16]. miRNA-122 has also been identified as a risk factor for hypertension, atherosclerosis, atrial fibrillation, acute myocardial infarction and heart failure[17–20]. In addition to miRNAs, emerging evidence revealed the important functions of new classes of sncRNAs including transfer RNA-derived small RNAs (tsRNAs) and ribosomal RNA-derived small RNA (rsRNAs) in regulating biological processes and disease development[21–29]. However, the functions of these "non-canonic" sncRNAs in lipid metabolism or CVD remain elusive.

As compared to the well-studied miRNAs, most tsRNAs and rsRNAs are highly modified and include various RNA modifications such as RNA methylations and terminal modifications[21,22,26,30,31]. When performing standard RNA sequencing (RNA-seq) analysis for these highly modified RNAs, it has been recognized that these modifications can interfere with the reverse transcription or adapter ligation processes during cDNA library preparation[21,32–34]. These conditions prevent the discovery of highly modified tsRNAs/rsRNAs that may have important functions in biological processes and disease development[21]. To overcome this obstacle, we recently developed an advanced PANDORA-seq method[22,35] which enable us and others to detect the highly modified sncRNAs that were otherwise undetectable by traditional RNA-seq protocol[21,22,36–40]. Intriguingly, PANDORA-seq unveiled tsRNA/rsRNA-enriched sncRNA landscapes in many murine and human tissues and cell types, which were strikingly different with traditional RNA-seq results[21,22,36–41].

In the present study, we identified tsRNA-Glu-CTC as the most abundant tsRNA in mouse liver by using PANDORA-seq. Interestingly, tsRNA-Glu-CTC is a cholesterol-responsive tsRNA that can regulate cholesterol homeostasis in vivo. While overexpression of tsRNA-Glu-CTC elicited hypercholesterolemia and hepatic steatosis in mice, antisense oligonucleotide (ASO)-mediated tsRNA-Glu-CTC knockdown protected mice from diet-induced hypercholesterolemia and atherosclerosis. Transcriptomic analysis revealed that tsRNA-Glu-CTC can regulate key hepatic genes including *Srebp2* that regulate lipid homeostasis. Mechanistic studies revealed that tsRNA-Glu-CTC can promote *Srebp2* transcription through an E-box motif in the promoter. RNA pulldown assay then demonstrated that tsRNA-Glu-CTC interacts with nuclear SREBP2 proteins to coordinately regulate *Srebp2* transcription via this E-box motif. Using a newly developed mass spectrometry-based direct sequencing approach[42], we then identified site-specific RNA modifications of endogenous tsRNA-Glu-CTC that are important for its function in regulating cholesterol homeostasis. Our findings reveal a previously unrecognized role of tsRNA in regulating cholesterol homeostasis and provide a potential new target for treating cardiometabolic disease.

## Results

### PANDORA-seq but not traditional small RNA-seq identifies tsRNA-Glu-CTC as the most abundant tsRNA in the liver

Many sncRNAs including tsRNAs and rsRNAs are highly modified and these RNA modifications can interfere with cDNA library construction process, preventing the detection of these sncRNAs by the widely used small RNA-seq methods[21,22]. To address this obstacle, we have recently developed an advanced PANDORA-seq to eliminate the RNA modification-elicited sequence interferences[21,22,35]. Liver is a key organ with numerous key functions including regulating lipid homeostasis.

To reveal the hepatic sncRNA profiles, we isolated total RNAs from mouse liver and performed both PANDORA-seq and traditional small RNA sequencing experiments. As expected, traditional RNA-seq identified miRNAs as the most abundant sncRNA species in the liver (Fig. 1a). However, PANDORA-seq revealed that liver contains more abundant tsRNAs and rsRNAs than miRNAs (Figs. 1a, b), which is consistent with our recent studies[22].

tsRNAs are among the most ancient small RNAs that presented in all three domains of life and numerous species[43,44]. tsRNAs have recently been demonstrated to have diverse roles in fundamental biological processes[41,43–47]. Interestingly, PANDORA-seq identified tsRNA-Glu-CTC as the most abundant hepatic tsRNA, which constitutes more than 65% of the total detected tsRNAs in the liver (Figs. 1c, d). Further, mapping tsRNA expression patterns on individual tRNA length scales demonstrated the advantages of PANDORA-seq in detecting tsRNA-Glu-CTC and several other tsRNAs including tsRNA-Asp-GTC, tsRNA-His-GTG, and tsRNA-Ser-GCT, as compared with traditional RNA-seq method (Fig. 1e). PANDORA-seq also detected many different tsRNA-Glu-CTC species but the highly expressed tsRNA-Glu-CTC species were mostly derived from the 5' end of tRNA-Glu-CTC (Fig. 1e and Supplementary Fig. 1).

We next compared the hepatic expression levels of tsRNA-Glu-CTC with well-studied hepatic miRNAs including miRNA-122 and miRNA-148a[13,14]. Previous studies demonstrated that these miRNAs are highly expressed in the liver[13,14]. For example, miRNA-122 has been shown to constitute 70% of the total miRNA pool in liver[13,14]. While traditional RNA-seq detected more miRNA-122 and miRNA-148a than tsRNA-Glu-CTC, PANDORA-seq demonstrated that tsRNA-Glu-CTC was more abundant than miRNA-122 or miRNA-148a in the liver (Fig. 1f). To confirm the PANDORA-seq results, we performed northern blot analysis for tsRNA-Glu-CTC and miRNA-122. Since northern blot analysis needs to use different probes to detect these sncRNAs, we first checked probe efficiency by loading the same amount of synthetic tsRNA-Glu-CTC and miRNA-122 oligonucleotides (Fig. 1g). Northern blot analysis then revealed that miRNA-122 probe has much higher hybridization efficiency than the tsRNA-Glu-CTC probe targeting 5' end of tsRNA-Glu-CTC (~4:1 based on densitometry analyses) (Fig. 1g). Next, we used these probes to evaluate hepatic miRNA-122 and tsRNA-Glu-CTC expression in male and female mice by norther blot analysis. As expected, miRNA-122 can be readily detected in mouse hepatic RNAs by norther blot using the highly efficient miRNA-122 probe (Fig. 1h). Despite lower probe efficiency, northern blot can also detect strong tsRNA-Glu-CTC bands in the same hepatic RNA samples (Fig. 1h). The quantification results based on densitometry analyses and appropriate normalizations confirmed that liver has relatively higher expression levels of tsRNA-Glu-CTC compared to miRNA-122 (~4:1) (Fig.1i). Therefore, PANDORA-seq but not traditional RNA-seq results represent in vivo situation of sncRNA expression. We next examined the tsRNA-Glu-CTC tissue expression pattern by northern blot. In addition to liver, tsRNA-Glu-CTC is expressed across major mouse organs including intestine, spleen, lung, and adipose tissue (Fig.1j). Taken together, these results confirmed PANDORA-seq's advantages in identifying the "hidden" sncRNAs such as tsRNA-Glu-CTC which are much more abundant than miRNA but are otherwise overlooked by traditional RNA-seq method.

### tsRNA-Glu-CTC is a cholesterol-responsive tsRNA

To explore the potential function of tsRNA-Glu-CTC in lipid metabolism, we used a classical cholesterol feeding approach[3,48] and fed the C57BL/6 wild-type (*WT*) mice with a semi-synthetic low-fat AIN76 diet (4.3% fat) containing either 0.02% or 0.5% cholesterol for 4 weeks[3,37,38,49]. Consistent with results from previous cholesterol feeding studies[3,48,50], feeding *WT* mice with the high-cholesterol diet (HCD) paradoxically did not affect circulating cholesterol or triglyceride levels but led to hepatomegaly and hepatic steatosis

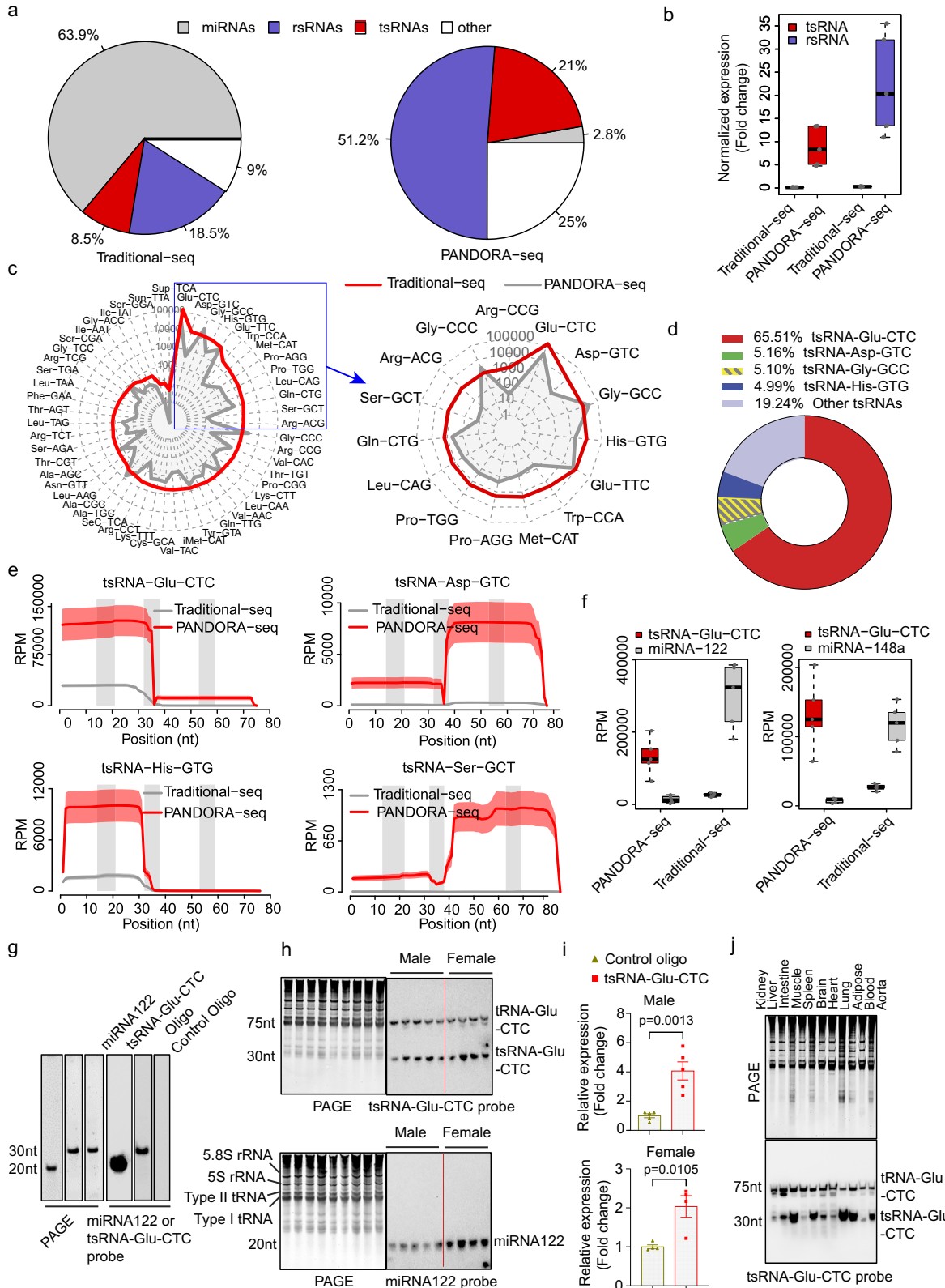

(Supplementary Fig. 2a–d). Hepatic cholesterol contents were also significantly increased in HCD-fed mice as compared to mice fed with the low-cholesterol diet (LCD) containing 0.02% cholesterol ($P = 0.0017$) (Supplementary Fig. 2e). As expected, gene expression analysis confirmed that high-cholesterol feeding suppressed hepatic expression of key lipogenic genes, including *Hmgcr* and *Psck9* but

stimulated several liver X receptor (LXR) target genes such as *Abcg5* and *Abcg8* (Supplementary Fig. 2f).

Intriguingly, both PANDORA-seq and traditional-seq analyses of hepatic sncRNAs revealed that HCD feeding significantly decreased tsRNA-Glu-CTC expression (Figs. 2a, b). Several other tsRNAs including tsRNA-Ser-GCT and tsRNA-Leu-CAG were not affected by HCD feeding

**Fig. 1 | PANDORA-seq identifies tsRNA-Glu-CTC as the most abundant hepatic tsRNA in mice. a** Distribution of different types of sncRNAs in mouse liver detected by traditional RNA-seq and PANDORA-seq (*n* = 5 biological replicates). The percentage of each sncRNA category was calculated by mean reads per million (RPM). **b** Hepatic tsRNA and rsRNA relative expression (normalized to miRNAs) under traditional-seq and PANDORA-seq protocols (*n* = 5 biological replicates). **c** The expression levels of highly expressed hepatic tsRNAs detected by traditional-seq (gray line) and PANDORA-seq (red line) are presented as the radar charts. The Y axis indicates RPM in log scale (*n* = 5 biological replicates). **d** The distribution of detected hepatic tsRNAs based on PANDORA-seq results (% of total hepatic tsRNAs) (*n* = 5 biological replicates). **e** Mapping patterns of representative hepatic tsRNAs detected by PANDORA-seq and traditional-seq. The solid curves indicate the mean RPM. The colored bands represent the standard error of the meaning (*n* = 5 biological replicates). The three gray bars indicate the location of the D-loop, anticodon-loop, and T-loop (left to right) along the parental tRNAs. **f** The expression levels of tsRNA-Glu-CTC, miRNA-122, and miRNA-148a based on PANDORA-seq and traditional-seq results (*n* = 5 biological replicates). **g** Hybridization efficiency of northern blot probes for tsRNA-Glu-CTC and miRNA-122 by loading 20 ng synthetic oligonucleotides. **h** Northern blot analysis of hepatic tsRNA-Glu-CTC and miRNA-122 in male and female mice. **i** Relative expression of hepatic miRNA-122 and tsRNA-Glu-CTC based on densitometry analyses of the northern blot bands normalized to indicated SYBR gold-stained RNA bands (5.8S rRNA, 5S rRNA, Type I and type II tRNA bands) on urea PAGE and the probe efficiency factor (4:1) (*n* = 5 male mice, *n* = 4 female mice, mean ± SEM, two-tailed Student's t-tests). **j** Northern blot analysis of tsRNA-Glu-CTC expression in major mouse tissues. For **g–j** similar results were obtained in three independent experiments. Box plots (**b, f**) show the median (central line), the 25th percentile (Q1) and 75th percentile (Q3) (bounds), and interquartile range (IQR, Q3-Q1) (box). The whiskers extend from the box to the most extreme data points that fall within Q3 + 1.5×IQR (upper whisker) and Q1-1.5×IQR (lower whisker), respectively.

(Fig. 2a). PANDORA-seq also identified many different tsRNA-Glu-CTC species that were downregulated in the liver by the HCD feeding (Fig. 2b) and most of them were derived from 5' end of tRNA-Glu-CTC which is consistent with the high expression levels of those tsRNA species (Fig. 1e and Supplementary Fig. 1). Northern blot analyses then validated PANDORA-seq results and demonstrated that hepatic expression of tsRNA-Glu-CTC but not tsRNA-Ser-GCT was downregulated by HCD feeding (Figs. 2c, d). Since HCD feeding led to decreased hepatic tsRNA-Glu-CTC expression in vivo, we next evaluated the impact of cholesterol environment on tsRNA-Glu-CTC expression in vitro. Human hepatic HepG2 cells were treated with acetylated-LDL (AcLDL) to mimic cholesterol loading conditions. Consistent with in vivo results, AcLDL treatment also decreased tsRNA-Glu-CTC expression in HepG2 cells (Fig. 2e). By contrast, depletion of cholesterol by simvastatin treatment led to increased tsRNA-Glu-CTC expression in those cells (Fig. 2f). Collectively, these results demonstrated tsRNA-Glu-CTC as a cholesterol-responsive tsRNA.

It is intriguing that high-cholesterol feeding led to decreased hepatic tsRNA-Glu-CTC expression. While tsRNA biogenesis has not been well understood, several tsRNA biogenesis-related enzymes inducing tsRNA cleavage enzymes and RNA modification enzymes have been known to regulate tsRNA biogenesis[38,41,43,44]. Next, we examined the hepatic expression of several key tsRNA biogenesis-related genes and found that HCD feeding led to significantly decreased mRNA levels of angiogenin (ANG), a key tsRNA cleavage enzyme in producing 5'tsRNA[43,47], in the live (Supplementary Fig. 2g). The expression levels of other tsRNA cleavage enzymes (e.g., RnaseT2 and RnaseL) and RNA modification enzymes (e.g., AlkB-1) were not affected by HCD feeding (Supplementary Fig. 2g). Consistently, exposure to acLDL also suppressed *Ang* expression in HepG2 cells in vitro (Supplementary Fig. 2h). By contrast, depletion of cholesterol by simvastatin treatment led to significantly increased ANG expression in those cells (Supplementary Fig. 2i). Therefore, it is plausible that the cholesterol environment affects the tsRNA biogenesis-related gene expression, leading to altered tsRNA-Glu-CTC expression.

### Treatment with synthetic tsRNA-Glu-CTC oligonucleotides leads to hypercholesterolemia and hepatic steatosis in mice

Since tsRNA-Glu-CTC is a cholesterol-responsive tsRNA, we next investigated whether it may affect cholesterol homeostasis in vivo. LCD-fed *WT* mice were treated with synthetic or control oligonucleotides every other day for two weeks by intraperitoneal (IP) delivery (Fig. 3a). As PANDORA-seq detected many different tsRNA-Glu-CTC species (Fig. 1e and Supplementary Fig. 1), we first selected a 30-nt tsRNA-Glu-CTC with medium expression levels as compared with other species (Supplementary Fig. 1). Northern blot analysis confirmed that IP delivery of synthetic 30-nt tsRNA-Glu-CTC oligonucleotides

(tsRNA-Glu-CTC Oligo-1) can successfully increase hepatic tsRNA-Glu-CTC expression in vivo (Fig.3b). tsRNA-Glu-CTC Oligo-1 treatment did not affect body weight (Fig. 3c) but led to significantly elevated serum total cholesterol levels and unchanged triglyceride levels in mice (Fig. 3d). Lipoprotein fraction analysis then revealed that tsRNA-Glu-CTC oligonucleotide treatment significantly increased serum LDL cholesterol levels without affecting VLDL or HDL cholesterol levels (Fig. 3e). Oil-Red-O and hematoxylin and eosin (H&E) staining of liver sections showed that tsRNA-Glu-CTC oligonucleotide-treated mice had increased lipid accumulation or hepatic steatosis (Fig. 3f). Consistently, tsRNA-Glu-CTC oligonucleotide treatment led to significantly increased hepatic cholesterol contents but did not affect hepatic triglyceride contents as compared to control mice (Fig. 3g). Furthermore, trichome staining results indicated that tsRNA-Glu-CTC overexpression also increased liver fibrosis (Supplementary Fig. 3a). Consistently, quantitative real-time PCR (QPCR) analysis demonstrated that tsRNA-Glu-CTC stimulated the expression of several key fibrotic genes including *Col1a1* and *Tgfβ* (Supplementary Fig. 3b).

In addition to the 30-nt tsRNA-Glu-CTC, PANDORA-seq detected a longer, 35-nt tsRNA-Glu-CTC which is highly expressed in the liver (Supplementary Fig. 1). We next treated mice with the synthetic 35-nt tsRNA-Glu-CTC oligonucleotides (tsRNA-Glu-CTC Oligo-2). The results showed the 35 nt tsRNA-Glu-CTC-2 oligonucleotides can elicit similar in vivo effects in mice including elevated serum cholesterol levels and hepatic steatosis (Supplementary Fig. 4a–d). Since both tsRNA-Glu-CTC Oligo-1 (30 nt) and tsRNA-Glu-CTC Oligo-2 (35 nt) elicited similar effects in vivo, the 5 extra nucleotides located at the 3' end of tsRNA-Glu-CTC Oligo-2 may not play important role in tsRNA-Glu-CTC-elicited lipogenic effects in vivo.

### Antisense oligonucleotide-mediated tsRNA-Glu-CTC knockdown improves lipid profile and reverses hepatic steatosis in high-cholesterol diet-fed mice

To further evaluate the potential role of tsRNA-Glu-CTC in cholesterol homeostasis, we used the antisense oligonucleotide (ASO) containing locked-nucleic acids (LNA) approach[51,52] to knock down tsRNA-Glu-CTC expression in vivo. *WT* mice were fed an HCD and then treated with tsRNA-Glu-CTC ASO by IP delivery for 2 weeks (Fig. 4a). Northern blot analysis confirmed that tsRNA-Glu-CTC ASO treatment led to reduced hepatic tsRNA-Glu-CTC expression without affecting mature tRNA-Glu-CTC levels (Figs. 4b, c). tsRNA-Glu-CTC ASO treatment did not affect body weight of those mice (Fig. 4d). Mice treated with tsRNA-Glu-CTC ASO had significantly decreased serum total cholesterol levels but unchanged triglyceride levels (Fig. 4e). The decreased total cholesterol levels were primarily due to reduced LDL cholesterol levels as HDL and VLDL cholesterol levels were not significantly affected by tsRNA-Glu-CTC ASO treatment (Fig. 4f). As expected, HCD feeding led to lipid accumulation in the

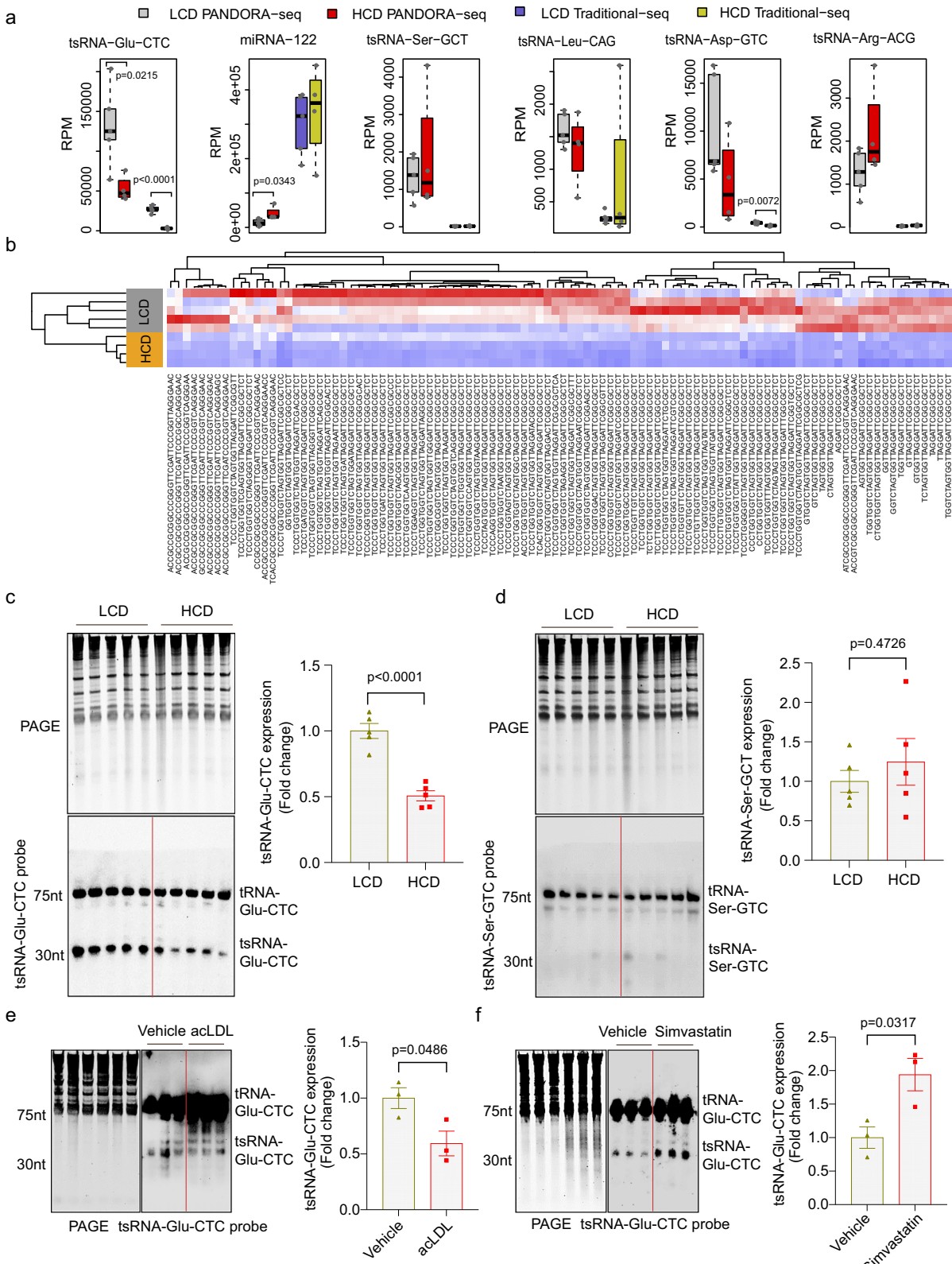

**Nature Communications** | (2025)16:11043

liver, but tsRNA-Glu-CTC ASO treatment ameliorated diet-induced hepatic steatosis (Fig. 4g). Consistently, mice exposed to tsRNA-Glu-CTC ASO had significantly reduced hepatic cholesterol contents, but similar triglyceride levels compared to mice treated with control ASO or vehicle control (Fig. 4h). Taken together, these results demonstrated the protective effects of tsRNA-Glu-CTC ASO against diet-induced hepatic steatosis.

## tsRNA-Glu-CTC regulates key hepatic lipogenic gene expression in mice

To elucidate the potential mechanisms for tsRNA-Glu-CTC oligonucleotide-elicited hypercholesterolemia and hepatic steatosis, hepatic RNAs were isolated for RNA sequencing (RNA-seq) analysis. RNA-seq analysis revealed that tsRNA-Glu-CTC overexpression led to 449 differentially expressed genes (DEGs) in mouse liver including 186

**Fig. 2 | tsRNA-Glu-CTC is a cholesterol-responsive tsRNA. a–d** Eight-week-old male wide-type C57BL/6 mice were fed a low-cholesterol diet (LCD) containing 0.02% cholesterol or high-cholesterol diet (HCD) containing 0.5% cholesterol for 4 weeks. Total RNAs were isolated from liver for RNA sequencing and northern blot analysis. **a** The expression levels of representatives tsRNAs or miRNAs based on PANDORA-seq and traditional-seq results ($n = 5$ mice for LCD, $n = 4$ mice for HCD, two-tailed Student's t-tests for HCD vs. LCD results obtained from PANDORA-seq and traditional-seq). Box plots show the median (central line), 25th percentile (Q1) and 75th percentile (Q3) (bounds), and interquartile range (IQR, Q3-Q1) (box). The whiskers extend from the box to the most extreme data points that fall within $Q3 + 1.5 \times IQR$ (upper whisker) and $Q1-1.5 \times IQR$ (lower whisker), respectively. **b** Heatmap representation of differentially expressed hepatic tsRNA-Glu-CTCs and their sequences ($n = 5$ mice for LCD, $n = 4$ mice for HCD). The top 100 tsRNA-Glu-CTC species based on Log-transformed counts per million (Log$^{CPM}$) were included. **c, d** Northern blot analyses of hepatic expression of tsRNA-Glu-CTC (**c**) and tsRNA-Ser-GCT (**d**) in LCD and HCD-fed mice ($n = 5$ biological replicates). The densitometry analyses of northern blot bands (normalized to SYBR gold-stained RNA bands, 5.8S rRNA, 5S rRNA, Type I and type II tRNA bands on urea PAGE) were shown next to northern blot panels (mean ± SEM, two-tailed Student's t-test). **e, f** Northern blot analysis of tsRNA-Glu-CTC expression in human hepatic cells, HepG2 cells, treated with vehicle control, acetylated LDL (25 ug/ml) (**e**) or simvastatin (5 uM) (**f**). The densitometry analysis of northern blot bands (normalized to SYBR gold-stained RNA bands on urea PAGE) were shown next to northern blot panels ($n = 3$ biological replicates, mean ± SEM, two-tailed Student's t-test). For **c–f** similar results were obtained in three independent experiments.

upregulated genes and 263 downregulated genes (Fig. 5a and Supplementary Table 1). Strikingly, Gene Ontology Biological Process (GOBP) analysis[53,54] revealed that tsRNA-Glu-CTC-stimulated genes were enriched in biological processes related to lipid homeostasis with "Cholesterol Biosynthetic Process" as the top altered GOBP term (Fig. 5b). Several other significantly upregulated GOBP terms included "Lipid Biosynthetic Process", "SREBP Signaling Pathways", and "Cholesterol Metabolic Process" (Fig. 5b). Consistently, the genes associated with these GOBP terms were upregulated in the liver of mice treated with tsRNA-Glu-CTC oligonucleotides as shown in the heatmap (Fig. 5c). We next performed Functional Analysis of Individual Microarray Expression (FAIME) algorithm to evaluate the geneset scores of the GOBP terms[37,38,54]. FAIME results further confirmed that tsRNA-Glu-CTC oligonucleotide treatment led to upregulation of geneset scores of those GOBP terms associated with lipid homeostasis in mouse liver (Fig. 5d).

Among those altered hepatic pathways and genes, we found it very intriguing that overexpression of tsRNA-Glu-CTC significantly upregulated "Cholesterol Biosynthetic Process" and "SREBP signaling pathway" and the associated *Srebp* genes such as *Srebp2*, a master lipogenic gene mediating cholesterol homeostasis (Figs. 5b–d). QPCR analysis validated RNA-seq results and demonstrated that tsRNA-Glu-CTC oligonucleotide treatment led to increased expression of *Srebp2* and its target genes *Hmgcr* and *Pcsk9* (Fig. 5e). Further, overexpression of tsRNA-Glu-CTC also stimulated *Srebp1c* and *Scd1* involved in fatty acid synthesis (Fig. 5e). Western blot analysis confirmed that both full-length, precursor form of SREBP2 (pSREBP2) and active form or nuclear SREBP2 (nSREBP2) protein levels were upregulated by tsRNA-Glu-CTC oligonucleotide treatment (Figs. 5f, g). Consistently, tsRNA-Glu-CTC oligonucleotide treatment also increased protein levels of HMGCR and PCSK9 (Figs. 5f, g). However, immunoblotting results showed that tsRNA-Glu-CTC overexpression did not affect the protein levels of several other lipid homeostasis-related molecules including Insig1, Insig2, SCAP, and SR-BI (Supplementary Fig. 5a, b). We also found that tsRNA-Glu-CTC overexpression did not affect the expression of several key intestinal lipogenic genes including *Npc1l1* and *Mttp* (Supplementary Fig. 5c).

Next, we analyzed hepatic lipogenic gene expression in mice treated with tsRNA-Glu-CTC ASO. We found that tsRNA-Glu-CTC ASO treatment led to significantly decreased *Srebp2* expression (Supplementary Fig. 6). tsRNA-Glu-CTC ASO treatment did not affect *Srebp1a* expression but reduced the expression levels of *Srebp1c* and *Fasn*. In addition, the expression levels of SREBP2 target genes including *Hmgcr* and *Pcsk9* were also downregulated by tsRNA-Glu-CTC ASO treatment (Supplementary Fig. 6). Collectively, these results demonstrate that tsRNA-Glu-CTC can alter key hepatic lipogenic gene expression and affect lipid homeostasis in mice.

## tsRNA-Glu-CTC regulates SREBP2 transcription through an E-box motif of its promoter

Animal experimental results indicate tsRNA-Glu-CTC oligonucleotide or ASO treatment can affect cholesterol homeostasis in vivo. It is intriguing

that overexpression of tsRNA-Glu-CTC increased the expression levels of SREBP2 and its target genes, but knockdown of tsRNA-Glu-CTC led to reduced expression of SREBP2. SREBP2 is a master regulator of cholesterol metabolism and can transcriptionally regulate many key genes including *Hmgcr* and *Pcsk9*. Unlike miRNAs, tsRNAs have more complex functions beyond RNAi-based mechanism[41,44] and previous studies indicated that tsRNAs can also promote gene transcription[41,44,55]. It is plausible that tsRNA-Glu-CTC may play a role in the transcriptional regulation of *Srebp2*. To test this hypothesis, we first isolated cytoplasmic and nuclear RNAs from hepatic HepG2 cells (Supplementary Fig. 7a) and found that tsRNA-Glu-CTC can be detected in both cytoplasm and nucleus contents by northern blot analysis (Fig. 6a). RNA fluorescence in situ hybridization (RNA-FISH) assays also confirmed that tsRNA-Glu-CTC can be found in the nucleus of HepG2 cells (Fig. 6b)

We next explored whether tsRNA-Glu-CTC can regulate SREBP2 transcription by luciferase reporter assays. A luciferase reporter vector containing 1.4 kb of *Srebp2* promoters was used for the transfection assays[56]. Interestingly, transfection of tsRNA-Glu-CTC oligonucleotides but not control oligonucleotides led to significantly increased luciferase reporter activities in HepG2 cells (Fig. 6c and Supplementary Fig. 7b). Next, we decreased tsRNA-Glu-CTC expression in HepG2 cells by transfection of tsRNA-Glu-CTC ASO (Supplementary Fig. 7c). The cells were also transfected with *Srebp2* luciferase reporters and treated with vehicle control or simvastatin. As expected, simvastatin treatment significantly increased *Srebp2* reporter activities (Fig. 7d). However, tsRNA-Glu-CTC ASO transfection reduced simvastatin-stimulated *Srebp2* reporter activities (Fig. 7d). These results suggest that modulation of tsRNA-Glu-CTC expression can regulate *Srebp2* transcription activities in vitro.

Several key promoter motifs including E-box, SRE-1 and NF-Y have been identified within the 1.4 kb *Srebp2* promoter region (Fig. 7e)[56,57]. We then constructed a series of promoter reporter vectors by inserting different lengths of *Srebp2* promoter regions into the luciferase reporter plasmids to identify potential tsRNA-Glu-CTC response elements in the *Srebp2* promoter region (Fig. 7e). Transfection with these reporter vectors showed that the vectors containing an E-box element can respond to tsRNA-Glu-CTC oligonucleotide treatment that led to significantly increased luciferase activities (Fig. 7f). However, tsRNA-Glu-CTC oligonucleotides had no effects on the luciferase activities of reporter plasmids that did not contain the E-box element (Fig. 7f). Next, promoter reporter vectors containing the wild-type *Srebp2* promoter or mutated E-box element were used to determine whether the E-box motif is necessary in mediating tsRNA-Glu-CTC transactivation (Fig. 4g). Consistently, tsRNA-Glu-CTC oligonucleotide transfection increased wild-type *Srebp2* reporter gene activities, but E-box mutant vectors did not respond to tsRNA-Glu-CTC oligonucleotide transfection (Fig. 4g). Collectively, these results suggest that E-box motif is required for tsRNA-Glu-CTC-mediated *Srebp2* transcription activities.

It is intriguing that tsRNA-Glu-CTC regulates *Srebp2* transcription activities through the E-box motif. We next used biotin-labeled tsRNA-Glu-CTC oligonucleotides to pull down tsRNA-Glu-CTC- crosslinked

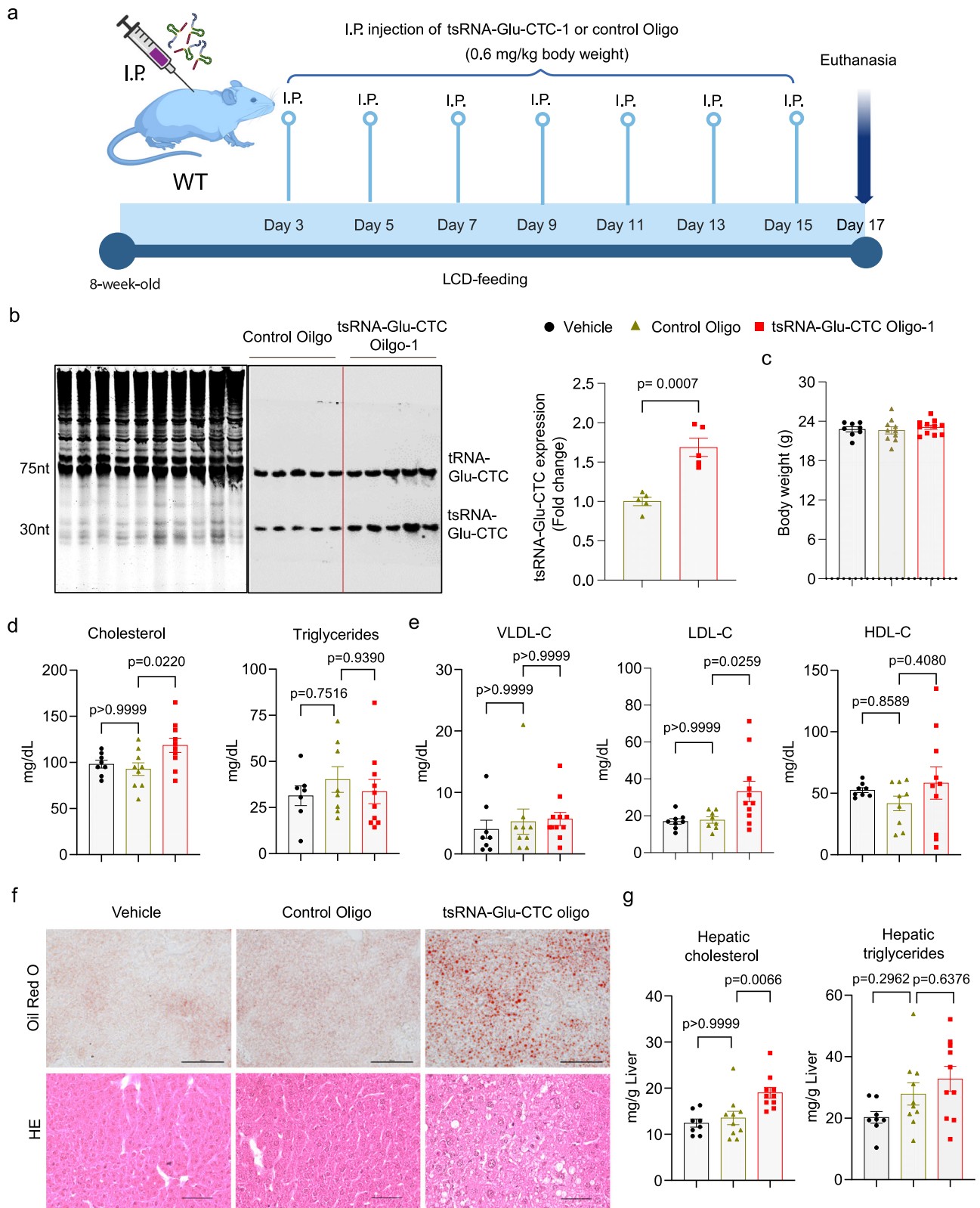

DNA fragments in HepG2 cells (Fig. 7h). The DNA fragments from biotin-labeled tsRNA-Glu-CTC pulldown samples were purified for PCR analysis (Fig. 7h). Specific primers targeting *Srebp2* promoter region containing the E-box motif were used for PCR analysis. We also used primers targeting promoters of several other genes including *Mttp* and *Cyp3a4* as controls[58,59]. Intriguingly, we found that tsRNA-Glu-CTC oligonucleotides can indeed pull down DNA fragments from the promoter region of *Srebp2* but not that of other genes such as *Mttp* and

*Cyp3a4* (Fig. 7i). Collectively, these results indicate that tsRNA-Glu-CTC can enter the nucleus and act through the E-box motif of *Srebp2* promoter to increase its transcription.

## tsRNA-Glu-CTC interacts with SREBP2 proteins to transcriptionally regulate its own expression

tsRNAs have been proposed to function as aptamers to interact with various RNA-binding proteins to exert their biological activities or

**Fig. 3 | tsRNA-Glu-CTC oligonucleotides elicit hypercholesterolemia and hepatic steatosis in mice.** Eight-week-old male C57BL/6 wild-type mice were treated with 0.6 mg/kg body weight synthetic 30-nt tsRNA-Glu-CTC oligonucleotides, control oligonucleotides, or vehicle control by intraperitoneal (IP) injection once every two days for two weeks before euthanasia. **a** The schematic of synthetic tsRNA-Glu-CTC oligonucleotide treatment experiment in wild-type mice. Created in BioRender.com. Zhou, C. (https://BioRender.com/hye39si). **b** Northern blot analysis of hepatic tsRNA-Glu-CTC expression in mice treated with control or tsRNA-Glu-CTC oligonucleotides. The densitometry analyses of northern blot bands (normalized to SYBR gold-stained RNA bands on urea PAGE) were shown next to northern blot panels ($n = 5$ biological replicates, mean ± SEM, two-tailed Student's t-test). Body weight ($n = 8,10,11$) (**c**), serum total cholesterol (left panel) ($n = 8,9,10$) and triglyceride (right panel) ($n = 7,8,10$) (**d**), cholesterol levels of lipoprotein fractions (VLDL-C, LDL-C, and HDL-C) ($n = 8,9,10$) (**e**), representative Oil-Red-O (top) and hematoxylin and eosin (bottom) stained liver sections (scale bar =100 μm) (**f**), and hepatic cholesterol (left panel) and triglyceride (right panel) contents ($n = 8,10,10$) (**g**) of mice treated with vehicle control, control oligonucleotides, or tsRNA-Glu-CTC oligonucleotides. $n$ represents the number of biological replicates (mice). Data are shown as mean ± SEM (one-way ANOVA, Bonferroni multiple-comparison test). *VLDL-C* very low-density lipoprotein cholesterol; *LDL-C* low density lipoprotein cholesterol; *HDL-C* high density lipoprotein cholesterol.

other RNAs[41,44]. As tsRNA-Glu-CTC regulates *Srebp2* transcription through an E-box motif, tsRNA-Glu-CTC may interact with nuclear proteins (e.g., transcription factors) involved in *Srebp2* transcription regulations. Intriguingly, SREBP proteins have been known to bind to E-box motifs to regulate target gene expression[6,60,61]. To test whether tsRNA-Glu-CTC can interact with SREBP2 proteins to facilitate its own transcriptional activities, we designed a biotin-labeled tsRNA-Glu-CTC probe which has the complementary oligonucleotide sequence matching tsRNA-Glu-CTC. We then used this probe to pull down tsRNA-Glu-CTC crosslinked proteins in HepG2 cells for immunoblotting analysis (Fig. 6j). Intriguingly, we found that the biotin-labeled tsRNA-Glu-CTC probe but not control probe can successfully pull down nSREBP2 proteins (Fig. 6k), indicating tsRNA-Glu-CTC may directly or indirectly interact with nSREBP2. Next, chromatin immunoprecipitation (ChIP) assays demonstrated that SREBP2 protein can indeed be recruited onto its own promotor region containing an E-box motif but not control promoters without the E-box motif in HepG2 cells (Fig. 6l).

We next performed reporter assays to confirm that transfection with *Srebp2* expression plasmids significantly increased the luciferase activities of wild-type *Srebp2* promoter but not the E-box mutant promoter (Fig. 6m). To investigate whether tsRNA-Glu-CTC regulates *Srebp2*'s transcriptional activities, similar reporter assays were performed by using the wild-type *Srebp2* promoter plasmids, *Srebp2* expression plasmids, and tsRNA-Glu-CTC ASO or control ASO (Fig. 6n). Knockdown of tsRNA-Glu-CTC by ASO decreased SREBP2-stimulated reporter activities (Fig. 6n), indicating the involvement of tsRNA-Glu-CTC in SREBP2-mediated transcriptional regulation. Consistent with tsRNA-Glu-CTC transfection assay results (Figs. 6c, f), tsRNA-Glu-CTC oligonucleotide transfection led to increased wild-type *Srebp2* promoter activities but shRNA-mediated *Srebp2* knockdown (Supplementary Fig. 7d) abolished tsRNA-Glu-CTC-stimulated *Srebp2* promoter activities (Fig. 6o). Taken together, these results demonstrate that tsRNA-Glu-CTC interacts with nSREBP2 proteins to coordinately regulate *Srebp2* transcription via the E-box motif.

### tsRNA-Glu-CTC ASO protects *Ldr receptor*-deficient mice from diet-induced hypercholesterolemia and atherosclerosis

Since tsRNA-Glu-CTC ASO treatment led to improved lipid profiles in HCD-fed *WT* mice, we next investigated whether knockdown of tsRNA-Glu-CTC can affect atherosclerosis development in atherosclerosis-prone *Ldl receptor*-deficient (*Ldlr-/-*) mice. *Ldlr-/-* mice fed a normal chow diet (ND) or HCD were treated with control ASO or tsRNA-Glu-CTC ASO (Fig. 7a). Consistent with previous studies[37,49], feeding *Ldlr-/-* mice with the HCD led to severe hypercholesterolemia with significantly elevated serum total, LDL, and VLDL cholesterol levels without affecting the body weight (Figs. 7b–e). tsRNA-Glu-CTC ASO treatment substantially ameliorated HCD-elicited hypercholesterolemia as mice treated with tsRNA-Glu-CTC ASO had significantly decreased serum total, LDL, VLDL cholesterol levels as compared with control ASO-treated mice (Figs. 7b–e). Interestingly, tsRNA-Glu-CTC ASO treatment also led to significantly decreased serum triglycerides levels as compared to control ASO-treated mice (Fig. 7d). HCD feeding stimulated

lipid accumulation and hepatic steatosis in *Ldlr-/-* mice but tsRNA-Glu-CTC ASO treatment ameliorated diet-induced hepatic steatosis in those mice (Fig. 7f). Consistently, tsRNA-Glu-CTC ASO treatment caused decreased hepatic cholesterol contents in *Ldlr-/-* mice (Fig. 7g).

Atherosclerotic lesion areas were then analyzed at the aortic roots of *Ldlr-/-* mice. As expected, HCD feeding led to substantial increased atherosclerotic lesion sizes at the aortic roots of *ldlr-/-* mice but tsRNA-Glu-CTC ASO treatment significantly decreased HCD-exacerbated atherosclerotic lesions in *Ldlr-/-* mice (Fig. 7h). The changed lesions size is likely driven by serum cholesterol levels as there was a positive correlation between atherosclerotic lesions sizes and serum cholesterol levels in those mice (Fig. 7i). In addition to lesion sizes, HCD-fed *Ldlr-/-* mice also had increased macrophage contents in the atherosclerotic plaques but tsRNA-Glu-CTC ASO treatment led to decreased macrophage contents in those mice (Fig. 5j). We next analyzed the hepatic expression of key lipogenic genes. Consistent with results obtained from *WT* mice, *Ldlr-/-* mice treated with tsRNA-Glu-CTC ASO had significantly decreased hepatic expression of key cholesterol homeostasis-related genes including *Srebp2*, *Hmgcr*, and *Hmgcs1* compared with control ASO-treated mice (Fig. 5k). Collectively, these results demonstrate that chronic tsRNA-Glu-CTC ASO treatment can ameliorate diet-induced hypercholesterolemia and atherosclerosis in *Ldlr-/-* mice.

### Circulating tsRNA-Glu-CTC levels are associated with increased cholesterol levels in humans

While our in vivo results suggest that tsRNA-Glu-CTC can regulate cholesterol homeostasis and atherosclerosis in mouse models, it is not clear whether tsRNA-Glu-CTC is associated with cholesterol levels in humans. In addition to mouse tissues, we found that tsRNA-Glu-CTC can also be detected in human liver and whole blood samples (Fig. 8a). In peripheral blood, tsRNA-Glu-CTC is mainly present in blood clot contents and can barely detected in serum by norther blot analysis (Fig. 8a). To explore the potential association between tsRNA-Glu-CTC and cholesterol metabolism in humans, a cohort of healthy human subjects were recruited for blood collection (Fig. 8b). Whole blood samples were used for serum and blood clot separation. Northern blot analysis was used to detect tsRNA-Glu-CTC levels in blood clot contents, and cholesterol levels were measured in the serum samples. We then found a positive correlation between tsRNA-Glu-CTC expression and serum cholesterol contents (Figs. 8c, d). Although these are only correlation results in a relatively small cohort, our findings provide the evidence that tsRNA-Glu-CTC expression is associated with serum cholesterol levels in humans.

### Identification and mapping of multiple RNA modifications of endogenous tsRNA-Glu-CTC

As compared to well-studied miRNAs, tsRNAs are often highly modified since their precursors, tRNAs have a wide variety of modifications such RNA methylations (e.g., $m^1G$, $m^1A$ and $m^3C$) which have important implications in tsRNA properties and function[21,22,26,30,31]. Since only PANDORA-seq but not traditional RNA-seq revealed that tsRNA-Glu-CTC is much more abundant than other well studied miRNAs in the

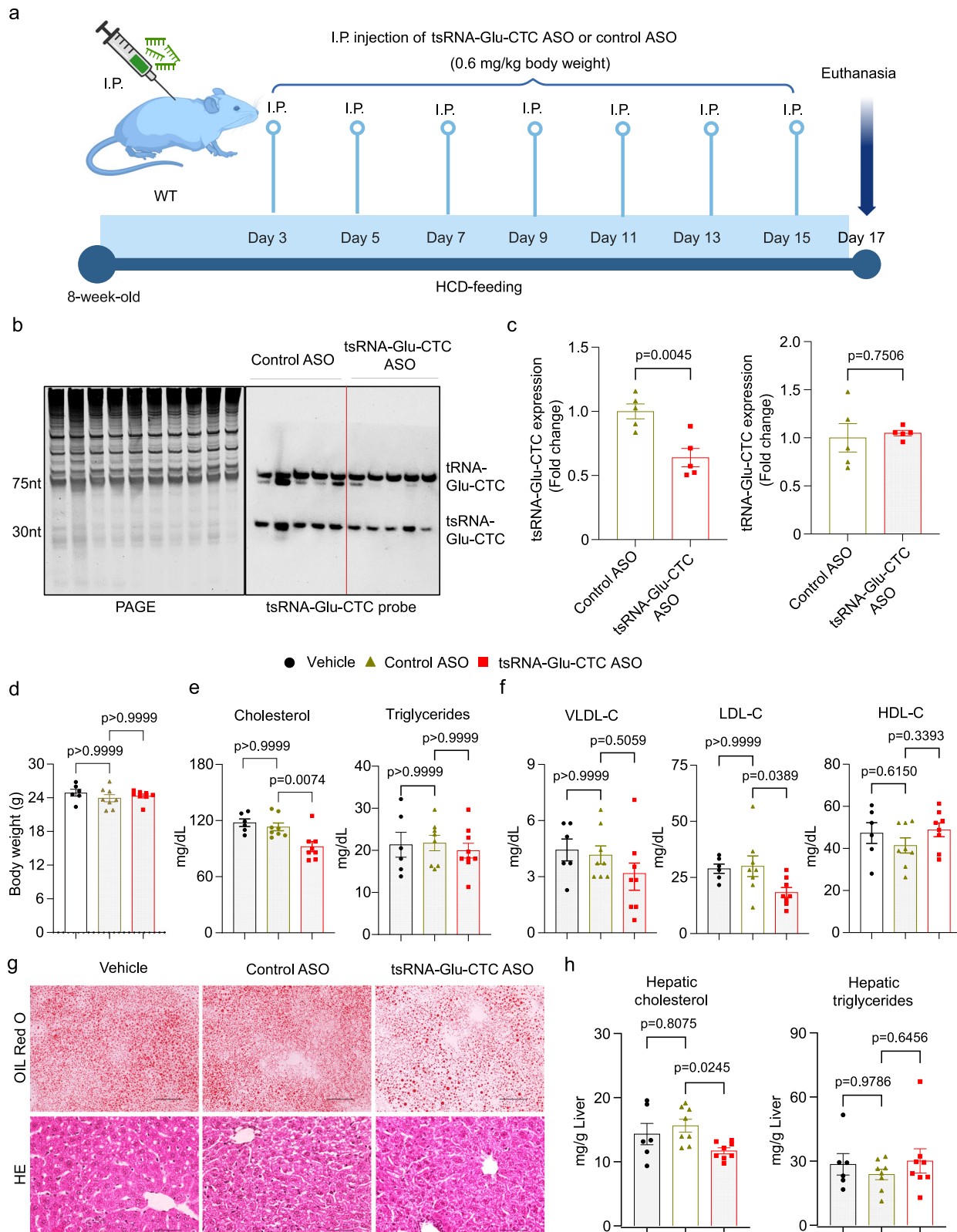

liver (Fig. 1), it is likely that tsRNA-Glu-CTC has specific RNA modifications that interfere with the cDNA library construction process.

To identify the potential RNA modifications of tsRNA-Glu-CTC, we first isolated endogenous tsRNA-Glu-CTC from mouse liver using a biotin-labeled tsRNA-Glu-CTC probe (Supplementary Fig. 8a). Next, we analyzed the RNA modifications of tsRNA-Glu-CTC by using a recently developed mass spectrometry (MS) ladder complementation

sequencing approach (MLC-seq) which allows quantitatively mapping tRNA nucleotide modifications site-specifically[21,42]. As a result, we detected 100% methylation of G at nucleotide 6 and 100% dihydrouridine (D) modifications at nucleotides 19 and 20 of the tsRNA-Glu-CTC (Figs. 9a–c and Supplementary Data 1). To further distinguish the isomeric methylated guanosine(G) such as m[1]G vs. m[2]G, we applied AlkB enzyme to treat tsRNA-Glu-CTC since AlkB can remove the methyl

**Fig. 4 | Knockdown of tsRNA-Glu-CTC by antisense oligonucleotides decreases circulating cholesterol levels and ameliorates hepatic steatosis in mice fed a high-cholesterol diet.** Eight-week-old male wild-type mice were fed a high-cholesterol diet (HCD) and were treated with tsRNA-Glu-CTC antisense oligonucleotides (ASO) or control ASO once every two days for two weeks. **a** The schematic of tsRNA-Glu-CTC ASO treatment experiment in wild-type mice. Created in BioRender.com. Zhou, C. (https://BioRender.com/ub1p0f6). **b** Northern blot analysis of hepatic tsRNA-Glu-CTC expression in mice treated with control ASO or tsRNA-Glu-CTC ASO (n = 5 biological replicates). **c** The densitometry analyses of northern blot bands for tsRNA-Glu-CTC and tRNA-Glu-CTC (normalized to SYBR gold-stained RNA bands on urea PAGE) (n = 5 biological replicates, mean ± SEM, two-tailed Student's t-test). Body weight (**d**), serum total cholesterol (left panel) and triglyceride (right panel) (left) (**e**), cholesterol levels of lipoprotein fractions (VLDL-C, LDL-C, and HDL-C) (**f**), representative Oil-Red-O (top) and hematoxylin and eosin (bottom) stained liver sections (scale bar =100 μm) (**g**), and hepatic cholesterol (left panel) and triglyceride (right panel) contents (**h**) of mice treated with vehicle control, control ASO, or tsRNA-Glu-CTC ASO (n = 6 for vehicle, n = 8 for control ASO, n = 8 for tsRNA-Glu-CTC ASO, data are shown as mean ± SEM, one-way ANOVA, Bonferroni multiple-comparison test). *VLDL-C* very low-density lipoprotein cholesterol; *LDL-C* low density lipoprotein cholesterol; *HDL-C* high density lipoprotein cholesterol.

group of $m^1G$, $m^1A$, and $m^3C$ and convert them to their respective canonical nucleotides[42]. However, AlkB is inert toward $m^2G$, $m^6A$, and $m^5C$[42]. The MLC-seq results of AlkB-treated tsRNA-Glu-CTC still showed 100% methylation of G at nucleotide 6 (Fig. 9b), confirming that the methylated G at this nucleotide is $m^2G$ (Fig. 9c).

In addition to tsRNA-Glu-CTC, we also isolated its parental tRNA, tRNA-Glu-CTC for RNA modification analysis. Consistent with tsRNA-Glu-CTC results, MLC-seq detected the same modifications at 5'end of tRNA-Glu-CTC including $m^2G$ at nucleotide 6 and D at nucleotides 19 and 20 (Supplementary Fig. 8b–d and Supplementary Data 1). Further, MLC-seq also identified other modifications located at the 3' half of tRNA-Glu-CTC including 100% $m^5C$ at nucleotides 48 and 49, 100% $m^5Um$ (5,2'-O-dimethyluridine) at nucleotide 23, and partial $m^1A$ at nucleotide 58 (Supplementary Fig. 8b–d). Since mC modification at nucleotides 48 and 49 were not affected by AlkB treatment, the specific modification is likely to be $m^5C$ (Supplementary Fig. 8c). By contrast, nucleotide 58 of the tRNA-Glu-CTC had a 33:67 ratio of mA to A, but this stoichiometry decreased to 13% mA after AlkB treatment (Supplementary Fig. 8c–e), suggesting it is likely $m^1A$, but not $m^6A$ (Supplementary Fig. 8e). Collectively, these results revealed the diverse RNA modifications of tsRNA-Glu-CTC and suggest that the unexplored features of these modifications may affect the function of tsRNA-Glu-CTC. Further, our study also demonstrated the advantage of MLC-seq in site-specifically mapping tRNA/tsRNA nucleotide modifications.

### tsRNA-Glu-CTC modifications are important for its function in the regulation of cholesterol homeostasis in vivo

RNA modifications may be crucial for regulating tsRNA properties including RNA stability and functions[43,44,62]. While mice treated with synthetic tsRNA-Glu-CTC oligonucleotides developed hypercholesterolemia and hepatic steatosis (Fig. 3), it is unknow whether endogenous tsRNA-Glu-CTC with modifications has similar or more potent effects in mice. To determine this, we first isolated endogenous tsRNA-Glu-CTC from *WT* mouse liver (Supplementary Fig. 8a). Due to the limited amount of endogenous tsRNA we isolated, we used a low dose of tsRNA-Glu-CTC (0.006 mg/kg body weight) to treat mice (Fig. 9d), which was 100 times lower than the dose used for synthetic oligonucleotide treatment (0.6 mg/kg body weight) (Fig. 3). In addition, we also included two groups of mice treated with the same low dose (0.006 mg/kg body weight) of synthetic tsRNA-Glu-CTC oligonucleotides or control oligonucleotides (Fig. 9d). Remarkably, the low dose of endogenous tsRNA-Glu-CTC treatment led to significantly increased serum total and LDL cholesterol levels in mice without affecting the body weight (Figs. 9e–g). By contrast, treatment of mice with the same dose of synthetic tsRNA-Glu-CTC oligonucleotides had no effects on the serum lipid profiles (Figs. 9f and 9g). Endogenous tsRNA-Glu-CTC treatment also led to increased lipid accumulation in the liver and significantly elevated hepatic cholesterol but not triglyceride contents (Figs. 9h, i). While Oil-Red-O staining indicated that the low dose of synthetic tsRNA-Glu-CTC oligonucleotides tended to increase lipid accumulation in the liver (Fig. 9h), the hepatic cholesterol and triglyceride contents were not significantly affected by the treatment with

0.006 mg/kg body weight synthetic tsRNA-Glu-CTC oligonucleotides (Fig. 9i). Exposure to endogenous tsRNA-Glu-CTC also led to significantly increased expression levels of key hepatic lipogenic genes including *Srebp2*, *Hmgcr*, and *Hmgcs1* (Fig. 9j). Western blot results also confirmed that endogenous tsRNA-Glu-CTC treatment increased the protein levels of full-length and active form of SREBP2, HMGCR, and PCSK9 (Fig. 9k). Taking together, these data demonstrated that endogenous tsRNA-Glu-CTC with corelated RNA modifications has much more potent effects on cholesterol homeostasis in vivo as compared with unmodified tsRNA-Glu-CTC oligonucleotides.

## Discussion

sncRNAs play diverse roles in regulating numerous biological processes in both normal and pathological conditions[63,64]. For CVD research, most studies have focused on investigating the function of miRNAs, previously considered as the dominant regulatory sncRNAs. These studies have led to the identification of important miRNAs in the regulation of lipid metabolism and CVD[4,7–11,65,66]. With advanced RNA sequencing methods, recent studies have unveiled surprisingly new sncRNA landscapes in various tissues and cells[21,41,44]. For example, we and others have recently demonstrated that many murine and human tissues or cells are enriched with numerous evolutionarily conserved sncRNAs derived from longer structure of parental RNAs (e.g., tRNAs, rRNAs, small nucleolar RNA)[21,22,36–41,44]. In the current study, we consistently found that tsRNAs and rsRNAs are much more abundant than the well-studied miRNAs in mouse liver. We then identify tsRNA-Glu-CTC as the most abundant hepatic tsRNA in mice. Interestingly, tsRNA-Glu-CTC is a cholesterol-responsive hepatic tsRNA, and gain-of-function and loss-of-function studies demonstrated that tsRNA-Glu-CTC can regulate cholesterol homeostasis in vivo (Fig. 10). While overexpression of tsRNA-Glu-CTC led to elevated circulating cholesterol levels and increased hepatic steatosis in mice, knockdown of tsRNA-Glu-CTC protected mice for diet-induced hypercholesterolemia and atherosclerosis development. Unlike miRNAs with mainly RNAi-based function mode, tsRNA-Glu-CTC can increase rather than decrease the expression levels of key hepatic lipogenic genes including *Srebp2*. Interestingly, tsRNA-Glu-CTC can enter the nucleus and act through an E-box motif of *Srebp2* promote to increase its transcription. We further demonstrated that tsRNA-Glu-CTC interacts with nSREBP2 proteins to coordinately regulate *Srebp2* transcription. Similar to its parental tRNA, tsRNA-Glu-CTC is also highly modified, and these RNA modifications are important for its function in regulating cholesterol metabolism in vivo. Collectively, our study demonstrated the function of a highly expressed hepatic tsRNA in regulating lipid homeostasis and atherosclerosis development (Fig. 10).

Traditional small RNA sequencing protocols often generate miRNA-enriched sncRNA profiles in bio-samples since miRNAs are usually not highly modified and can be easily detected by these protocols[21,22,41]. The main cause of sequencing bias is derived from two main aspects during the cDNA library preparation: one is due to internal RNA modifications (e.g., $m^1G$, $m^1A$) that interfere with reverse transcription process and the other is due to terminal modifications (e.g., phosphate/cyclic phosphate at the 3'-termini) that prevent

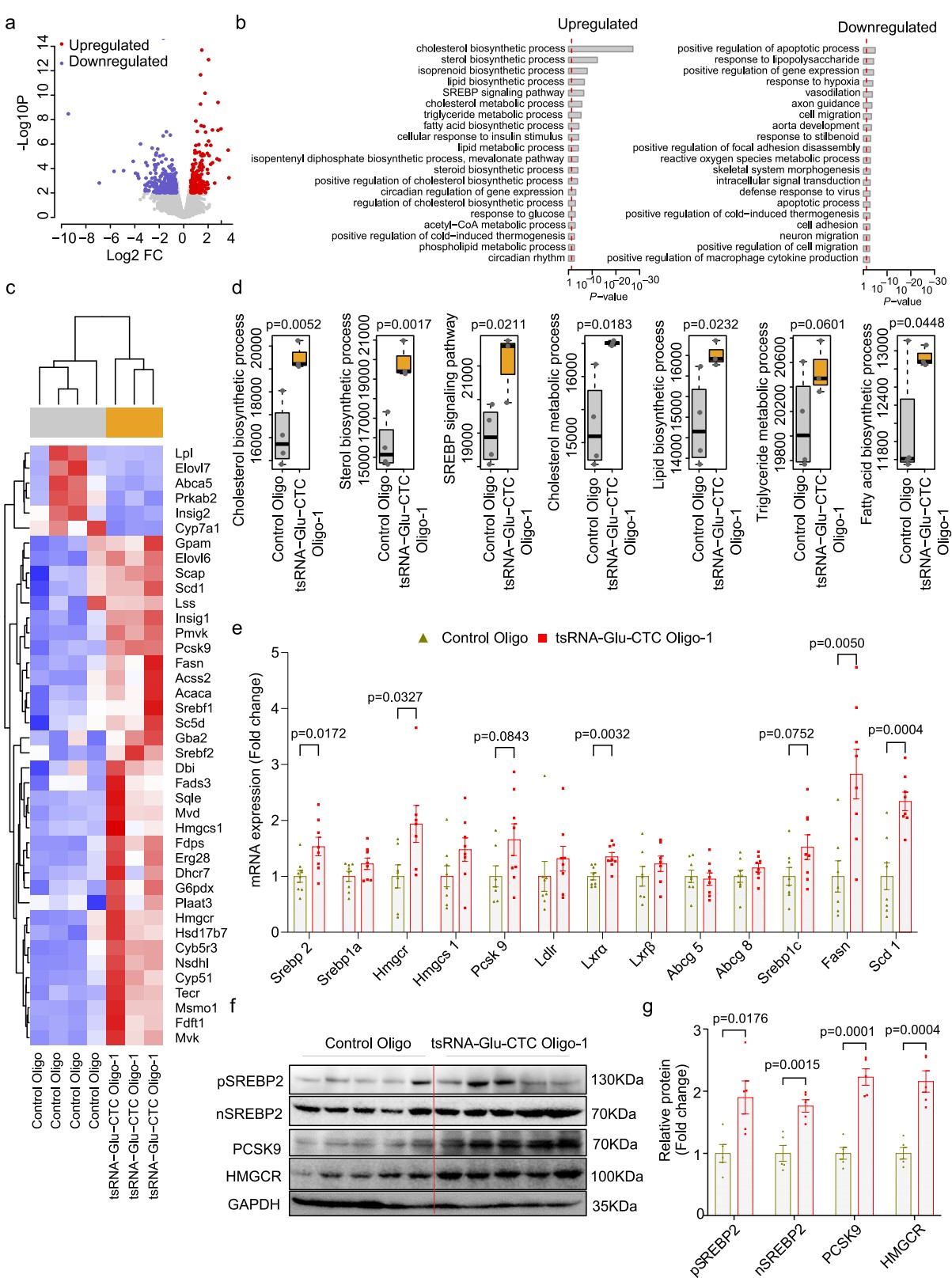

adapter ligation process[21,22]. Due to inherent limitations of the traditional sequencing methods, sncRNA results generated from these traditional methods may be biased and may not represent the true composition of the sncRNA profiles. These issues have been well-recognized by the field, and several methods have been developed to address these limitations including ARM-seq, CPA-seq, and AQRNA-seq[21,67]. Based on the previous work, we recently refined the

methodology and developed the PANDORA-seq protocol[22,35] which removes key RNA modifications that block adapter ligation and reverse transcription during cDNA library construction by step-wise enzymatic treatments[21,22,35]. PANDORA-seq method enabled us and others to identify highly modified sncRNAs and to unveil tsRNA/rsRNA-enriched sncRNA landscapes in many tissues and cells[21,22,36–39,41,68]. In the current study, we consistently found that mouse liver contains

**Fig. 5 | tsRNA-Glu-CTC regulates key hepatic lipogenic gene expression in mice.** Eight-week-old male wild-type mice were treated with 0.6 mg/kg body weight synthetic 30-nt tsRNA-Glu-CTC oligonucleotides or control oligonucleotides once every two days for two weeks. Total RNAs were isolated from the liver and used for RNA-seq analysis (**a**) Volcano plot of differential expressed genes (DEGs) in mouse liver. Colored dots show the upregulation (red dots) or downregulation (blue dots) of DEGs. The edgeR tool was applied to identify DEGs with $P$ value < 0.01 and a fold change (FC) > 1.5 as a cut-off threshold ($n = 4$ mice for control oligo, $n = 3$ mice for tsRNA-Glu-CTC oligo). Associated data are provided in Supplementary Table 1. **b** Gene Ontology Biological Process (GOBP) terms significantly associated with the hepatic DEGs. The $P$ values were one-sided and computed using the DAVID tool. Exact $P$ values are provided in the Source Data file. **c** Heatmap representation of DEGs involved in GOBP terms of "Cholesterol biosynthetic process", "Sterol biosynthetic process", "SREBP signaling pathway", "cholesterol metabolic process" and "lipid biosynthetic process". Each row indicates one individual gene and each column a biological replicate of mouse. Red represents upregulation of gene expression and blue indicates downregulation. **d** Geneset scores of the prioritized GOBP terms. The geneset score was calculated by the FAIME algorithm ($n = 4$ for control oligo, $n = 3$ for tsRNA-Glu-CTC oligo, one-tailed Student's t-tests). Box plots show the median (central line), the 25th percentile (Q1) and 75th percentile (Q3) (bounds), and interquartile range (IQR, Q3-Q1) (box). The whiskers extend from the box to the most extreme data points that fall within Q3 + 1.5×IQR (upper whisker) and Q1-1.5×IQR (lower whisker), respectively. **e** The expression levels of hepatic lipogenic genes of mice treated with control oligonucleotides or tsRNA-Glu-CTC oligonucleotides were analyzed by quantitative real-time PCR ($n = 8$ biological replicates, mean ± SEM, two-tailed Student's t-test). **f** Western blot analysis of precursor SREBPs (pSREBE2), nuclear SREBP (nSREBP), PCSK9, HMGCR, and GAPDH in the liver of mice treated with control oligonucleotides or tsRNA-Glu-CTC oligonucleotides ($n = 5$ biological replicates). **g** Densitometry analysis of western blot bands (normalized to GAPDH) ($n = 5$ biological replicates, mean ± SEM, two-tailed Student's t-test).

much more abundant tsRNAs/rsRNAs than miRNAs by using PANDORA-seq. We then identified tsRNA-Glu-CTC as the most abundant hepatic tsRNA, supported by northern blot results that hepatic expression levels of tsRNA-Glu-CTC were even higher than miRNA-122, the most abundant miRNA in the liver[13,14]. These results demonstrated the strengthens of PANDORA-seq in revealing more accurate sncRNA landscapes in bio-samples and in discovering the "hidden" sncRNAs that may have important functions in biological processes and disease development.

While previous studies revealed the important role of several miRNAs in cardiometabolic disease, the function of tsRNAs in lipid metabolism and CVD remain elusive. Using a classical cholesterol feeding approach[3,48], we found that tsRNA-Glu-CTC is a cholesterol-responsive hepatic tsRNA. The cholesterol feeding approach has contributed to the identification of important lipogenic genes including novel SREBP target genes including *Pcsk9* in previous mouse studies[3,69,70]. We then revealed that tsRNA-Glu-CTC can indeed affect lipid homeostasis in vivo. Treatment of synthetic tsRNA-Glu-CTC oligonucleotides can cause hypercholesterolemia and hepatic steatosis in *WT* mice, but ASO-mediated tsRNA-Glu-CTC knockdown reversed hypercholesterolemia and hepatic steatosis in HCD-fed mice. Similarly, tsRNA-Glu-CTC ASO protected atherosclerosis-prone *Ldlr*[-/-] mice from diet-induced hypercholesterolemia and atherosclerosis. RNA-seq analysis revealed that tsRNA-Glu-CTC regulated many hepatic genes regulating lipid homeostasis, and the most significantly upregulated GOBP term by tsRNA-Glu-CTC oligonucleotide treatment is "Cholesterol Biosynthetic Process" in mouse liver. Interestingly, the expression levels of SREBP2, the master regulator of cholesterol metabolism, were increased by tsRNA-Glu-CTC oligonucleotide treatment but decreased by tsRNA-Glu-CTC ASO treatment in both *WT* and *Ldlr*[-/-] mice, indicating SREBP2 as a potential target of tsRNA-Glu-CTC.

While HDL is the main circulating lipoproteins in *WT* mice, we found that overexpression or knockdown of tsRNA-Glu-CTC mainly affected serum LDL cholesterol levels but did not alter HDL cholesterol levels. Plasma cholesterol levels including HDL levels can be significantly changed in mouse models when SREBP2 pathway is altered by PCSK9[71,72]. The impact of PCSK9 overexpression or deletion on LDLR levels was substantial in those studies and overexpression of PCSK9 led to diminished hepatic LDLR expression[71]. In our study, the impact of tsRNA-Glu-CTC treatment or knockdown on hepatic lipogenic genes was relatively modest, which may lead to unchanged HDL levels. Previous studies from us and others also showed that altering certain lipogenic genes or pathways may only affect VLDL or LDL cholesterol levels without affecting HDL cholesterol levels in animal models with HDL as the main circulating lipoproteins[11,50,58,59,73–78]. For example, Trib1 has been demonstrated to regulate hepatic lipogenesis, and deficiency of Trib1 in *WT* mice led to increased plasma total cholesterol levels by 54%[73]. The increased cholesterol contents were predominantly mediated by significantly elevated VLDL and LDL cholesterol levels, but the HDL cholesterol levels remained unchanged[73]. Other groups also showed unchanged HDL cholesterol levels in mouse models for sncRNA-related studies. For example, miRNA-483 can target *Pcsk9*, leading to decreased hepatic PCSK9 protein and increased hepatic LDLR expression[11]. However, overexpression of miRNA-483 in *WT* mice only lowered circulating LDL and intermediate-density lipoprotein levels without affecting HDL levels[11]. Different circulating lipoprotein changes may also depend on experimental design, feeding conditions, and related genes or pathways. For example, ABCA1 is one of key molecules regulating HDL metabolism, and liver-specific deletion of *Abca1* can lead to over 80% decreased circulation HDL in mice[79]. Several well-studied miRNAs including miRNA-33 and miRNA-148 can target *Abca1* to regulate plasma HDL levels in mice[4,9,12,80,81]. In our studies, we found that overexpression or knockdown of tsRNA-Glu-CTC in vivo did not affect the expression or *Abca1* or other HDL metabolism-related genes (e.g., *Abcg1*, *Lxr*, *Abcg5/8*), which could also lead to unchanged HDL levels in vivo

Many non-coding RNAs including long non-coding RNAs (lncRNAs) and sncRNAs have been demonstrated to play important roles in regulating lipid metabolism. Several lncRNAs including LeXis, lncLSTR, and CHROME can regulate lipid homeostasis through different mechanisms[82–86]. For example, LeXis interacts with RNA-binding protein Raly to affect *Srebp2* transactional activity and to regulate cholesterol homeostasis and atherosclerosis in mice[82,83]. For sncRNAs, miRNA-33 coordinates with *Srebp* host genes to regulate cholesterol homeostasis in both mice and non-human primates[8–10]. miRNA-148a targets both *Ldlr* and *Abca1* to affect LDL and HDL levels in mice[4,12]. miRNA-30c and miRNA-483 can also reduce hyperlipidemia by targeting *Mtp* and *Pcsk9*, respectively[11,87]. These sncRNAs are well known to have RNAi-based functionality. tsRNA-Glu-CTC, however, increased rather than inhibited the expression of *Srebp2* in our study. Moreover, we also detected tsRNA-Glu-CTC in both cytoplasm and nucleus, and it can promote *Srebp2* transcription activities via an E-box motif. These results suggest that tsRNA-Glu-CTC exerts its function beyond RNAi-based mechanism. It has also been proposed that tsRNAs may exert their biological activities based on their secondary and tertiary structures[41,44]. While tRNAs can fold into complex structures which are essential for its functionality, tsRNAs derived from their precursors may have similar inherited structure features or new structures depending on the RNA length, modification status, and cleavage site[41,44]. The folded tsRNAs may function as aptamers to interact with various RNA-binding proteins or other RNAs. Thus, tsRNA-Glu-CTC may interact with nuclear proteins (e.g., transcription factors, histone modification enzymes) involved in *Srebp2* transcription regulation. Intriguingly, SREBPs have been known to bind to E-box motifs to regulate their target gene expression[6,60,61]. Our studies then suggested that both tsRNA-Glu-CTC and nSREBP2

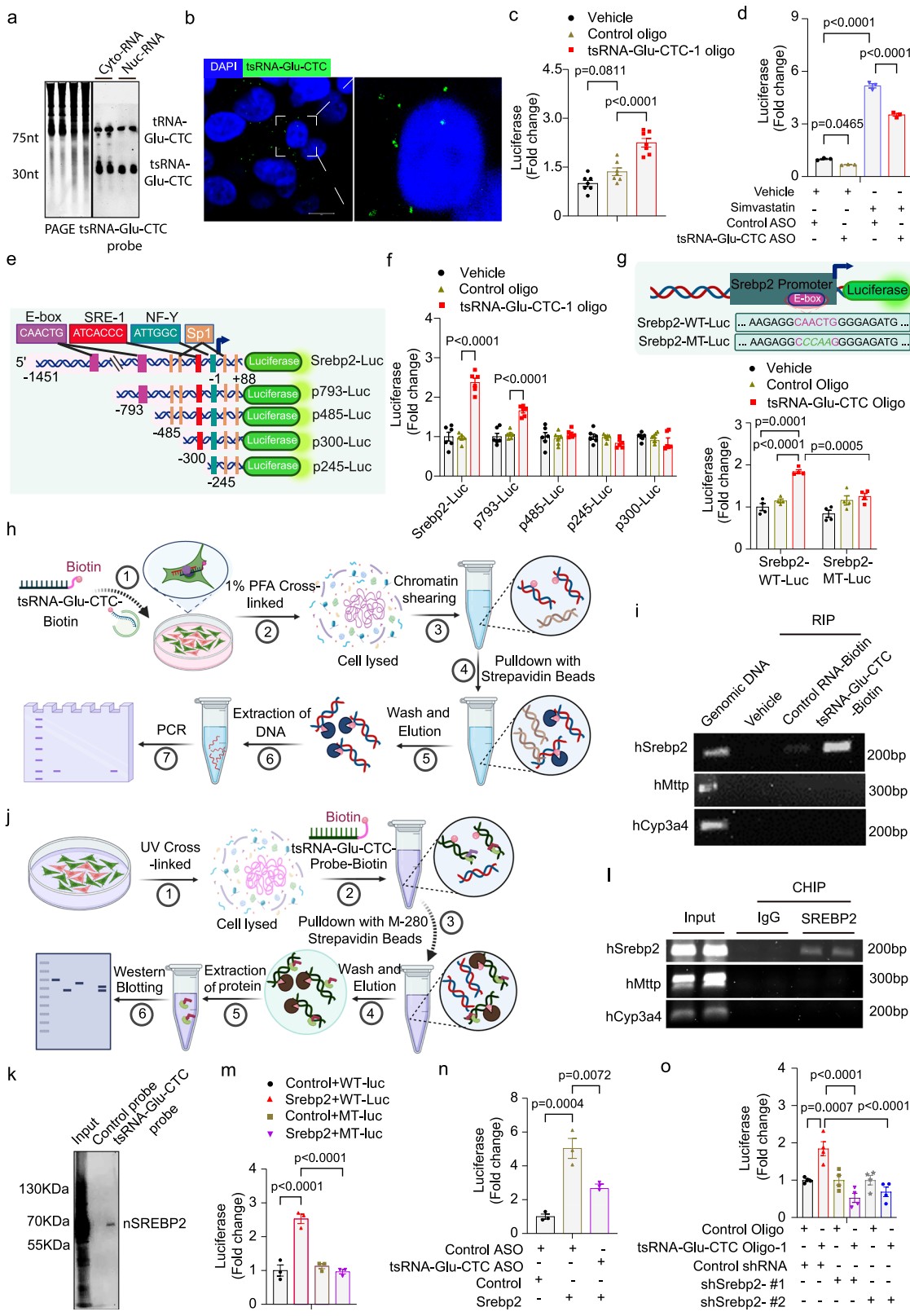

proteins can be recruited onto the E-box motif of *Srebp2* promoter. Biotin-labeled probe successfully pulled down nSREBP2 proteins as tsRNA-Glu-CTC associated proteins. Reporter assays further confirmed that tsRNA-Glu-CTC and nSREBP2 proteins coordinately regulated *Srebp2* transcription via the E-Box motif. While our results demonstrated the important new mechanism through which tsRNA-Glu-CTC regulates *Srebp2* expression, RNA-seq results showed overexpression of tsRNA-Glu-CTC altered the expression of many genes and pathways in the liver. Thus, it is very likely that tsRNA-Glu-CTC has complex functions in mediating lipid homeostasis by affecting multiple pathways or genes but not exclusively through *Srebp2*. Future studies are required to investigate the detailed mechanisms through which tsRNA-Glu-CTC regulate other genes and pathways to regulate lipid homeostasis in vivo.

**Fig. 6 | tsRNA-Glu-CTC interacts with SREBP2 to regulate its transcription through an E-box motif. a** Northern blot analysis of tsRNA-Glu-CTC in nuclear and cytoplasmic RNAs isolated from HepG2 cells. **b** Representative RNA-FISH analysis of the subcellular localization of tsRNA-Glu-CTC in HepG2 cells (scale bar =10 μm). **c** HepG2 cells were transfected with *Srebp2* promoter reporters and tsRNA-Glu-CTC oligonucleotides or control oligonucleotides together with CMX-*β-Galactosidase* control plasmids. Data are shown as fold induction of normalized luciferase activity compared with vehicle control group (*n* = 7 biological replicates, mean ± SEM, one-way ANOVA, Bonferroni multiple-comparisons test). **d** HepG2 cells were transfected with *Srebp2* promoter reporters and tsRNA-Glu-CTC ASO or control ASO. Cells were then treated with 5 μM simvastatin for 24 hr (*n* = 3, mean ± SEM, one-way ANOVA, Bonferroni multiple-comparison test). **e** The schematic of different *Srebp2* promoter reporter plasmids. Created in BioRender. Zhou, C. (https://BioRender.com/kjpasvb). **f** HepG2 cells were transfected with indicated *Srebp2* promoter reporters and tsRNA-Glu-CTC oligonucleotides or control oligonucleotides (*n* = 3, mean ± SEM, one-way ANOVA, Bonferroni multiple-comparison test). **g** The schematic of *Srebp2* promoter reporter plasmids containing E-box mutations (top panel). Created in BioRender. Zhou, C. (https://BioRender.com/7ybsw60). HepG2 cells were transfected with wild-type (*Srebp2*-WT-Luc) or mutant (*Srebp2*-MT-Luc)

reporters and tsRNA-Glu-CTC oligonucleotides or control oligonucleotides (*n* = 3, mean ± SEM, one-way ANOVA, Bonferroni multiple-comparison test). **h** The schematic of pulldown of tsRNA-Glu-CTC-associated DNA fragment experiment. Created in BioRender. Zhou, C. (https://BioRender.com/i0r477p). **i** PCR analysis of tsRNA-Glu-CTC-associated DNA fragments by using primers for indicated genes' promoters. **j** The schematic of pulldown of tsRNA-Glu-CTC-associated protein experiment. Created in BioRender. Zhou, C. (https://BioRender.com/2bnzfvk). **k** Western blot analysis of tsRNA-Glu-CTC-associated proteins in HepG2 cells. **l** ChIP analysis of the recruitment of SREBP2 onto the E-box region of *Srebp2* promoter. **m** HepG2 cells were transfected with *Srebp2*-WT-Luc or *Srebp2*-MT-Luc and SREBP2 expression or control plasmids (*n* = 3, mean ± SEM, one-way ANOVA, Bonferroni multiple-comparison test). **n** HepG2 cells were transfected with *Srebp2* promoter reporter, *Srebp2* expression or control plasmids, tsRNA-Glu-CTC ASO or control ASO (*n* = 3, mean ± SEM, one-way ANOVA, Bonferroni multiple-comparison test). **o** HepG2 cells were transfected with *Srebp2* promoter reporter plasmids, tsRNA-Glu-CTC oligonucleotides or control oligonucleotides, control shRNAs or two different shRNAs targeting *Srebp2* together with CMX-*β-Galactosidase* plasmids (*n* = 4, mean ± SEM, one-way ANOVA, Bonferroni multiple-comparison test). For **e**, **g**, **h**, and **i** the images were created with BioRender.com.

---

While PANDORA-seq can reveal the sequences of highly modified sncRNAs such as tsRNAs by removing the RNA modifications, these RNA modifications could be essential for their biological functions[41,44,62,88]. However, the currently available tools for detecting RNA modifications and their position are mainly designed for detecting a few well-known modifications such as m[1]A and m[5]C for long RNAs[21]. In the current study, we used an innovative LC-MS-based MLC-seq to map the RNA modifications of tsRNA-Glu-CTC and its precursor tRNA-Glu-CTC[42]. MLC-seq de novo sequences RNAs and their site-specific RNA modifications through the mass difference between the mass ladders[42], which are generated by time-controlled formic acid-mediated RNA degradation[21,89]. Compared with other traditional MS-based RNA methods that do not provide RNA modification positional information, MLC-seq has a big advantage by simultaneously mapping the RNA sequence and modifications of sncRNAs with single-nucleotide and stoichiometric precision[21,42]. Indeed, MLC-seq enabled us to map the RNA modifications of endogenous tsRNA-Glu-CTC and tRNA-Glu-CTC isolated from mouse liver. Interesting, we found that tsRNA-Glu-CTC has the same RNA modifications as its parental RNA, tRNA-Glu-CTC including m[2]G at nucleotide 6 and D at nucleotides 19 and 20, suggesting that tsRNA-Glu-CTC inherits these modifications from its precursor tRNA. More importantly, the endogenous modified tsRNA-Glu-CTC can regulate lipid homeostasis in vivo at a 100-time lower dose compared to its synthetic counterpart in our study (i.e., 0.006 mg/kg/BW vs. 0.6 mg/kg/BW). The modifications identified in this study may contribute to the high bioactivity of endogenous tsRNA-Glu-CTC, since RNA modifications have been proposed to module the basic properties of sncRNAs such as stability and the 2D/3D structure, and their physiological functions[44,62,90]. It would be interesting to investigate the detailed mechanisms through which the identified RNA modifications (m[2]G and D) affect tsRNA-Glu-CTC stability and functions.

While most investigations in the field still focus on the functions of "canonical" sncRNA such as miRNAs, emerging studies started to reveal the important function of tsRNAs in biological processes and disease development. For example, Goodarzi et al. previously reported a specific set of tsRNAs including tsRNA-Glu-YTC and tsRNA-Asp-GTC can suppress tumorigenesis by interacting oncogenic RNA-binding protein YBX1[51]. Kim et al. found that tsRNA-Leu-CAG regulates ribosome biogenesis and cell viability, and inhibition of tsRNA-Leu-CAG suppresses tumor growth[52]. tsRNA-Gln-CTG has been demonstrated to ameliorate liver injury[91], and tsRNA-His-GUG can activate Toll-like receptor 7 and regulate innate immune response[92]. Interestingly, a more recent study showed that tsRNA-Glu-CTC can be induced by

aging in the brain and it may regulate aging-related phenotypes in mice[93]. In the current study, we found that tsRNA-Glu-CTC is a cholesterol-responsive tsRNA in the liver that regulates lipid homeostasis in vivo. Thus, these tsRNAs have complex functions in different tissues. We also identified the specific modifications including m[2]G and D of tsRNA-Glu-CTC which may be critical for its function in lipid metabolism. ASO-mediated tsRNA-Glu-CTC inhibition can ameliorate diet-induced hyperlipidemia and atherosclerosis, suggesting tsRNA-Glu-CTC as a potential therapeutic target for cardiometabolic disease. Our results established tsRNA-Glu-CTC as a key regulator of lipid homeostasis. Findings from our studies will hopefully stimulate more investigations of the functions and mechanisms of understudied tsRNAs in biological processes and human diseases.

## Methods
### Animals
Wild-type (*WT*) C57BL/6 mice (Strain #000664) and *Ldlr*[-/-] mice on C57BL/6 background (Strain# 002207) were purchased from The Jackson Laboratory. Eight-week-old male and female *WT* mice were fed a semisynthetic low-fat (4.2% fat) AIN76 diet containing either low cholesterol (LCD; 0.02% cholesterol; Research Diets, D00110804C) or high cholesterol (HCD; 0.5% cholesterol; Research Diet, D00083101C) for 4 weeks before euthanasia and tissue collection for small RNA sequencing experiments and other experiments[37,59,78]. For tsRNA-Glu-CTC synthetic oligonucleotide treatment, 8-week-old male WT mice were fed the LCD and treated with 0.6 or 0.006 mg/kg body weight of 30-nt tsRNA-Glu-CTC oligonucleotides (tsRNA-Glu-CTC Oligo-1) or 35-nt tsRNA-Glu-CTC oligonucleotides (tsRNA-Glu-CTC Oligo-2) (Supplementary Table 2) by intraperitoneal injections every two days for two weeks before euthanasia. For endogenous tsRNA-Glu-CTC treatment, 8-week-old male *WT* mice were fed the LCD and treated with 0.006 mg/kg body weight of endogenous tsRNA-Glu-CTC or control small RNA isolated from mouse liver every two days for two weeks before euthanasia. For antisense oligonucleotide (ASO) treatment, 8-week-old male WT mice were fed a HCD and were treated with 0.6 mg/kg body weight of tsRNA-Glu-CTC ASO or control ASO (Supplementary Table 2) every two days for two weeks before euthanasia. tsRNA-Glu-CTC ASO has a complementary sequence of 19 nt of tsRNA-Glu-CTC (Supplementary Table 2). Both control ASO and tsRNA-Glu-CTC ASO contain locked nucleic acid (LNA) modifications to enhance binding affinity, increase nuclease resistance and improve potency[51,52,94]. ASOs used for this study were also phosphorothioate modified to increase their stability For atherosclerosis study, 7-week-old male *Ldlr*[-/-] mice were treated with 0.6 mg/kg body weight of tsRNA-Glu-CTC ASO or

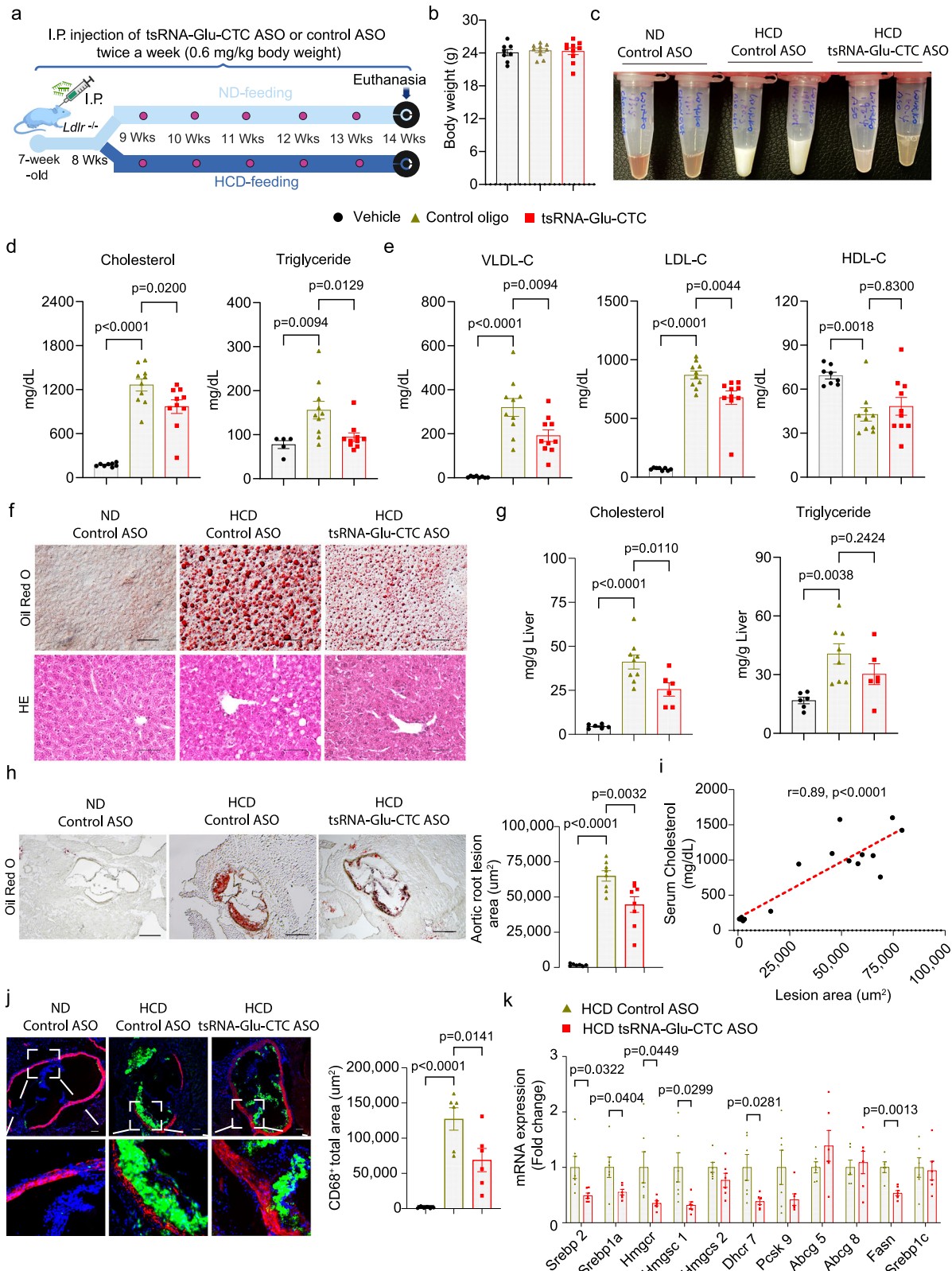

control ASO twice per week for 7 weeks before euthanasia. The mice were either fed a normal chow diet (ND) (LabDiet, PicoLab Rodent Diet 5053) for 7 weeks or fed a ND for 1 week and then switched to a HCD for 6 weeks. Mice were housed in microisolator cages and were provided ad libitum access to standard chow diet before the treatment and deionized water in temperature-controlled room ( ~ 21 °C) with 12 hr light/dark cycle and humidity ranging from 30-70%. On the day of euthanasia, mice were fasted for 6 hr following the dark cycle (feeding cycle) and blood and major organs were collected as previously described[36,37,77,78]. All animal studies were performed in compliance with the approved protocols by the Institutional Animal Care and Use Committee of the University of California, Riverside.

**Fig. 7 | tsRNA-Glu-CTC antisense oligonucleotides protects *Ldl receptor*-deficient mice from diet-induced hypercholesterolemia and atherosclerosis.**
Seven-week-old male *Ldl receptor*-deficient (*Ldlr⁻/⁻*) mice were treated with tsRNA-Glu-CTC antisense oligonucleotides (ASO) or control ASO twice per week for 7 weeks. The mice were maintained on a normal chow diet (ND) or switched to a high-cholesterol diet (HCD) starting at 8-week-old age. **a** The schematic of tsRNA-Glu-CTC ASO treatment experiment. Created in BioRender. Zhou, C. (https://BioRender.com/rj9jwzr). **b–g** Body weight (*n* = 8,10,10) (**b**), representative photos of collected serum (**c**), serum total cholesterol (*n* = 8,10,10) (left panel) and triglyceride (*n* = 5,10,10) (right panel) levels (**d**), cholesterol levels of lipoprotein fractions (VLDL-C, LDL-C, and HDL-C) (*n* = 8,10,10) (**e**), representative Oil-Red-O (top) and hematoxylin and eosin (bottom) stained liver sections (scale bar =100 μm) (**f**), and hepatic cholesterol (*n* = 6,9,6) (left panel) and triglyceride (*n* = 6,8,6) (right panel) contents (**g**) of *Ldlr⁻/⁻* mice treated with vehicle control, control ASO, or tsRNA-Glu-CTC ASO (data are shown as mean ± SEM, one-way ANOVA, Bonferroni

multiple-comparison test). **h** Representative images of Oil-red-O-stained sections and quantitative analysis of the atherosclerotic lesion area at the aortic root of *Ldlr⁻/⁻* mice (*n* = 7,8,8; mean ± SEM, one-way ANOVA, Bonferroni multiple-comparison test). **i** Correlation between serum total cholesterol levels and atherosclerotic lesion sizes in *Ldlr⁻/⁻* mice (*n* = 16; Correlation was analyzed by Pearson correlation). **j** Representative images of immunofluorescence staining of CD68 (green) and α-smooth muscle actin (αSMA) (red) at the aortic root of *Ldlr⁻/⁻* mice (scale bar=200 μm). Quantitative analysis of staining area is displayed as indicated (*n* = 6,6,6; mean ± SEM, one-way ANOVA, Bonferroni multiple-comparison test). **k** The expression levels of hepatic lipogenic genes of HCD-fed *Ldlr⁻/⁻* mice treated with tsRNA-Glu-CTC ASO or control ASO were analyzed by quantitative real-time PCR (*n* = 6, mean ± SEM, two-tailed Student's t-test). *n* represents the number of biological replicates (mice). *VLDL-C* very low-density lipoprotein cholesterol; *LDL-C* low density lipoprotein cholesterol; *HDL-C* high density lipoprotein cholesterol.

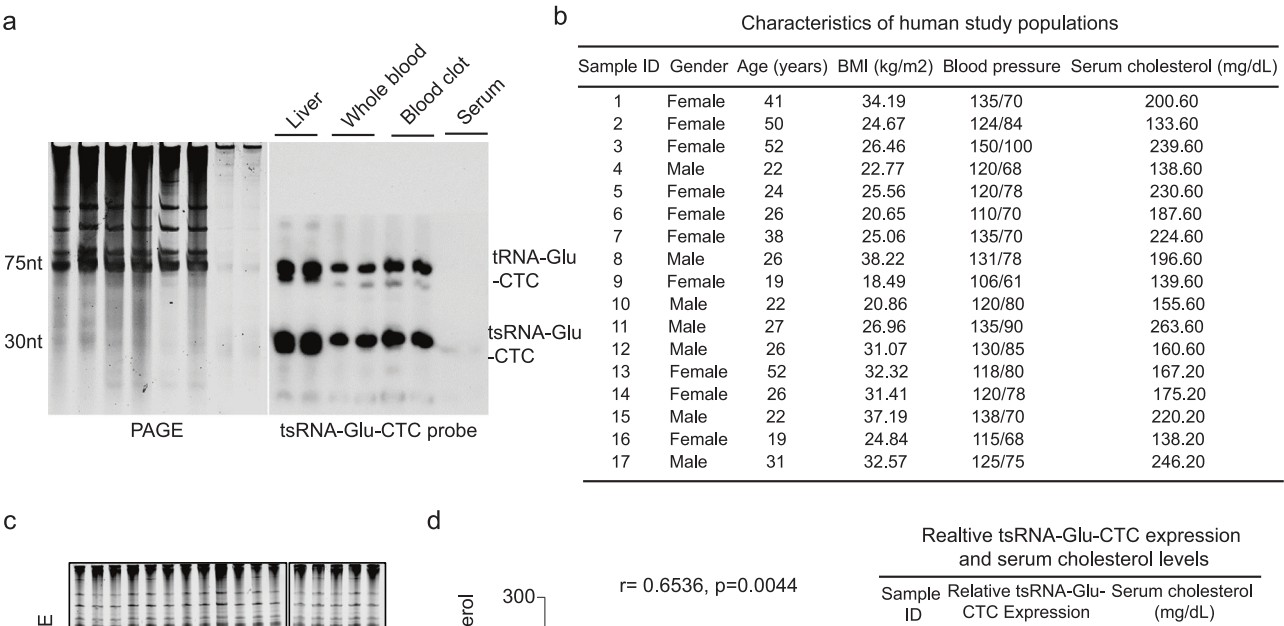

Characteristics of human study populations

| Sample ID | Gender | Age (years) | BMI (kg/m2) | Blood pressure | Serum cholesterol (mg/dL) |
|---|---|---|---|---|---|
| 1 | Female | 41 | 34.19 | 135/70 | 200.60 |
| 2 | Female | 50 | 24.67 | 124/84 | 133.60 |
| 3 | Female | 52 | 26.46 | 150/100 | 239.60 |
| 4 | Male | 22 | 22.77 | 120/68 | 138.60 |
| 5 | Female | 24 | 25.56 | 120/78 | 230.60 |
| 6 | Female | 26 | 20.65 | 110/70 | 187.60 |
| 7 | Female | 38 | 25.06 | 135/70 | 224.60 |
| 8 | Male | 26 | 38.22 | 131/78 | 196.60 |
| 9 | Female | 19 | 18.49 | 106/61 | 139.60 |
| 10 | Male | 22 | 20.86 | 120/80 | 155.60 |
| 11 | Male | 27 | 26.96 | 135/90 | 263.60 |
| 12 | Male | 26 | 31.07 | 130/85 | 160.60 |
| 13 | Female | 52 | 32.32 | 118/80 | 167.20 |
| 14 | Female | 26 | 31.41 | 120/78 | 175.20 |
| 15 | Male | 22 | 37.19 | 138/70 | 220.20 |
| 16 | Female | 19 | 24.84 | 115/68 | 138.20 |
| 17 | Male | 31 | 32.57 | 125/75 | 246.20 |

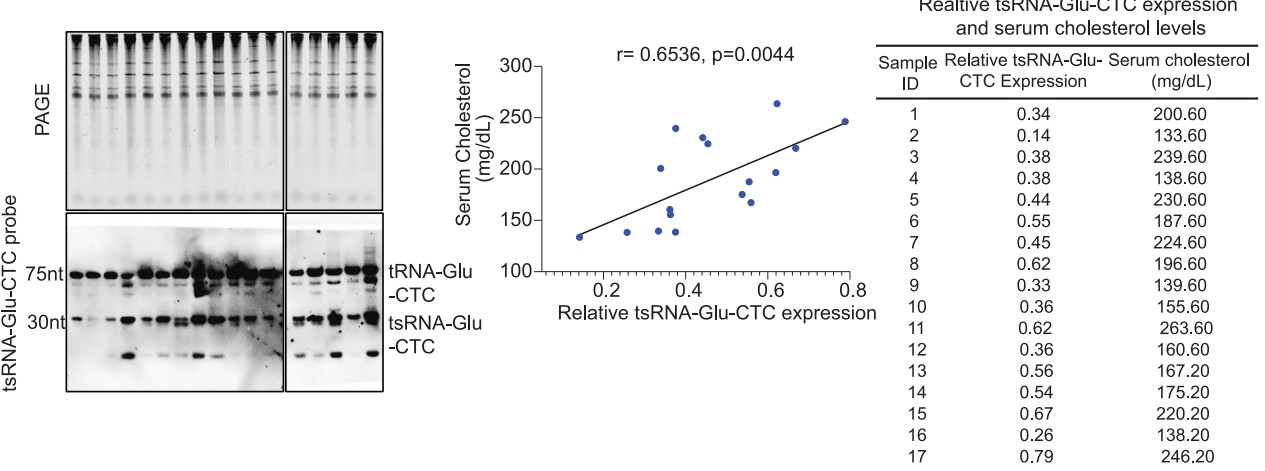

Realtive tsRNA-Glu-CTC expression and serum cholesterol levels

| Sample ID | Relative tsRNA-Glu-CTC Expression | Serum cholesterol (mg/dL) |
|---|---|---|
| 1 | 0.34 | 200.60 |
| 2 | 0.14 | 133.60 |
| 3 | 0.38 | 239.60 |
| 4 | 0.38 | 138.60 |
| 5 | 0.44 | 230.60 |
| 6 | 0.55 | 187.60 |
| 7 | 0.45 | 224.60 |
| 8 | 0.62 | 196.60 |
| 9 | 0.33 | 139.60 |
| 10 | 0.36 | 155.60 |
| 11 | 0.62 | 263.60 |
| 12 | 0.36 | 160.60 |
| 13 | 0.56 | 167.20 |
| 14 | 0.54 | 175.20 |
| 15 | 0.67 | 220.20 |
| 16 | 0.26 | 138.20 |
| 17 | 0.79 | 246.20 |

**Fig. 8 | Circulating tsRNA-Glu-CTC levels are correlated with serum cholesterol levels in humans. a** Representative northern blot analysis of tsRNA-Glu-CTC expression in human liver, whole blood, blood clot, and serum portion (similar results were obtained in three independent experiments). **b** Characteristics of individual human subjects. **c** Whole blood collected from human subjects were separated into serum and blood clot parts. Northern blot analysis of tsRNA-Glu-

CTC expression in blood clot contents of human subjects (*n* = 17). **d** Correlation between relative tsRNA-Glu-CTC expression levels based on the densitometry analyses of northern blot bands (normalized to SYBR gold-stained RNA bands on urea PAGE) and serum cholesterol levels. The data values were shown next to the correlation panel. The correlation was analyzed by Pearson correlation (*n* = 17). n represents the number of biological replicates.

## Human subjects

Human blood samples were collected under the protocol approved by the University of California, Riverside Clinical Institutional Review Board (HS 19-076). The study was performed in accordance with the *Declaration of Helsinki*, except for registration in a database. Consent procedures were performed in the participant's

native language with study personnel fluent in the language (English or Spanish). All participants were provided with a copy of the consent form prior to their appointment and were informed of the purpose of the study, including all risks and benefits. Participants provided written informed consent prior to participation and were monetarily compensated. Participants were recruited by word of

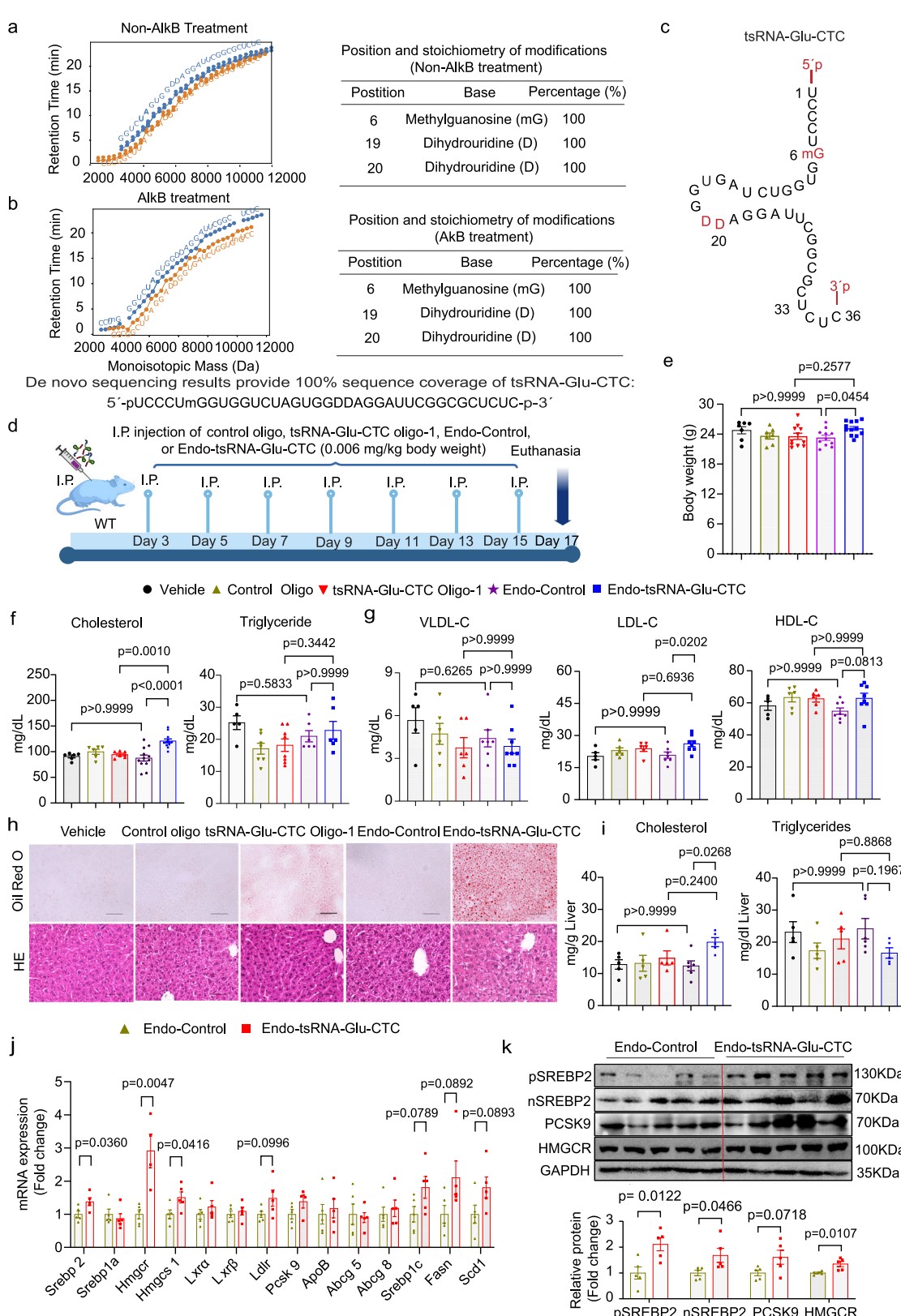

mouth and advertisements near the University of California, Riverside campus and local clinical sites. Inclusion criteria included age ≥18 years. Exclusion criteria included severe cardiac or pulmonary illness, confirmed or suspected active COVID-19 infection, and pregnancy. Participants were asked not to consume caffeine on the morning prior to the study and to abstain from taking anti-inflammatory medication, corticosteroids, or other medications that could potentially interfere with downstream measurements. The study included 17 healthy participants ($n = 10$ women, $n = 7$ men) between 19 to 52 years of age. Participants self-reported their sex at birth and current gender identity. The mean age was $25.14 \pm 1.28$ years old for men and $34.70 \pm 4.27$ years old for women, and body mass index was $29.94 \pm 2.54$ kg/m$^2$ for men and $26.36 \pm 1.58$ kg/m$^2$ for women (Fig. 8b).

**Fig. 9 | Endogenous tsRNA-Glu-CTC with RNA modifications elicits more potent effects on circulating and hepatic cholesterol levels in mice.** Endogenous tsRNA-Glu-CTC was isolated from the liver of wild-type mice. **a–c** MLC-seq of untreated (**a**) or AlkB-treated (**b**) endogenous hepatic tsRNA-Glu-CTC. Computational isolation of mass spectrometry (MS) data for all ladder fragments derived from the endogenous tsRNA-Glu-CTC in both the 5′-and 3′-ladders out of the complex MS data of mixed samples with multiple distinct RNA strands. Identity, position, and stoichiometry of each modification are listed in the tables (**a** and **b**) and tsRNA-Glu-CTC picture (**c**). mG: methylguanidine; D: dihydrouridine. MLC-seq associated data are included in Supplementary Data 1. **d–i** Eight-week-old male wild-type mice were treated with vehicle control (Vehicle), 0.006 mg/kg body weight of control oligonucleotides (Control Oligo), synthetic 30-nt tsRNA-Glu-CTC oligonucleotides (tsRNA-Glu-CTC Oligo-1), endogenous total small RNAs isolated from mouse liver (Endo-Control), or endogenous tsRNA-Glu-CTC isolated from mouse liver by intraperitoneal injection once every two days for two weeks before euthanasia. The schematic of endogenous tsRNA-Glu-CTC treatment experimental design (**d**).

Created in BioRender. Zhou, C. (https://BioRender.com/ma7fndc). Body weight ($n$ = 7,7,11,11,12) (**e**), serum total cholesterol ($n$ = 6,6,7,12,12) (left panel) and triglyceride ($n$ = 5,7,8,6,6) (right panel) (**f**), cholesterol levels of lipoprotein fractions including VLDL-C, LDL-C, and HDL-C ($n$ = 5,6,6,7,8) (**g**), representative Oil-Red-O (top) and hematoxylin and eosin (bottom) stained liver sections (scale bar =100 μm) (**h**), and hepatic cholesterol (top panel) and triglyceride (bottom panel) contents ($n$ = 5,5,5,5,6) (**i**) of treated mice. For **e**, **f**, **g**, and **i**, data are shown as mean ± SEM (one-way ANOVA, Bonferroni multiple-comparison test). **j** The expression levels of hepatic lipogenic genes of mice treated with Endo-Control or Endo-tsRNA-Glu-CTC were analyzed by quantitative real-time PCR ($n$ = 5, mean ± SEM, two-tailed Student's t-test). **k** Western blot analysis of hepatic SREBP2 (pSREBP2 and nSREBP2), PCSK9, and HMGCR protein levels. Densitometry analysis of western blot bands (normalized to GAPDH) was shown below western blot panels ($n$ = 5, mean ± SEM, two-tailed Student's t-test). For **e–k** $n$ represents the number of biological replicates (mice). *VLDL-C* very low-density lipoprotein cholesterol; *LDL-C* low density lipoprotein cholesterol; *HDL-C* high density lipoprotein cholesterol.

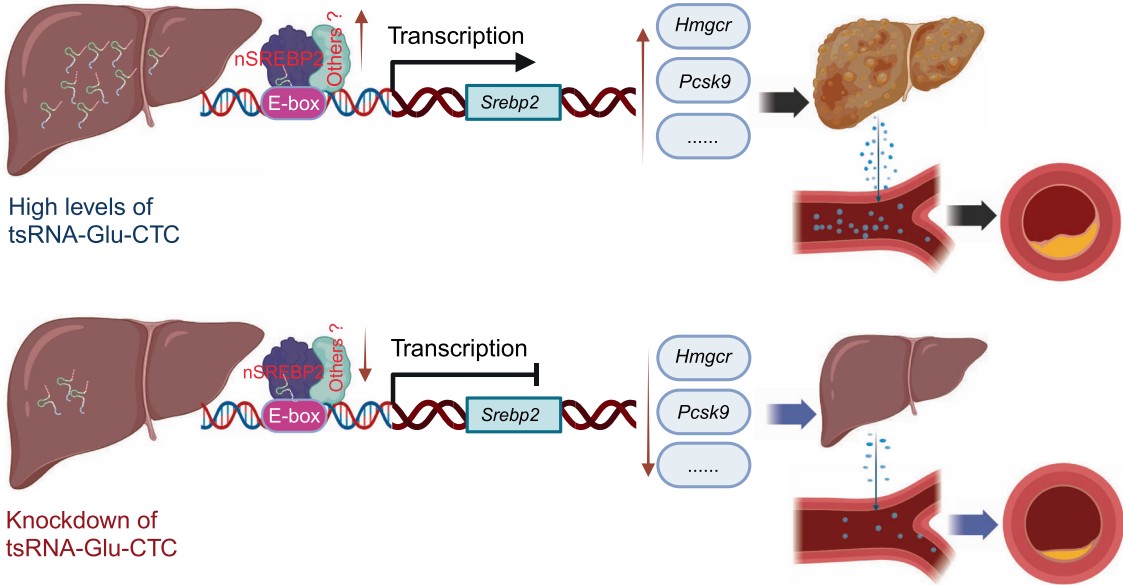

**Fig. 10 | Schematic of the function of tsRNA-Glu-CTC in regulating lipid homeostasis and atherosclerosis development.** tsRNA-Glu-CTC, the most abundant hepatic tsRNA, is identified as a cholesterol-responsive tsRNA. tsRNA-Glu-CTC can regulate the expression of key hepatic lipogenic genes including *Srebp2*, a master regulator of lipid metabolism. tsRNA-Glu-CTC interacts with SREBP2 to regulate its own transcription through an E-box motif of *Srebp2* promoter. Knockdown of tsRNA-Glu-CTC suppresses the expression of *Srebp2* and other lipogenic genes, leading to protective effects against diet-induced hypercholesterolemia and atherosclerosis. Created in BioRender. Zhou, C. (https://BioRender.com/ml9ngwr).

Peripheral venous blood was collected during fasting by a licensed phlebotomist using standard venipuncture procedures. Samples were originally collected as part of a related study by members of our team[95]. Samples have been conserved in the Heinrich lab. Twenty-five ml of blood was drawn, with 5 ml of whole blood collected in a serum vacutainer tube for quantification of serum cholesterol and tsRNA-Glu-CTC. Following collection, serum blood tubes were immediately placed on ice for transport from the collection site, then allowed to clot at room temperature before centrifugation to separate blood clot contents and serum. The serum cholesterol levels were measured by using the Wako Cholesterol E enzymatic colorimetric assay kit (Wako, 999-02601) as previously described[37,78]. Total RNAs were isolated from non-serum portion of whole blood using TRIzol reagents (Sigma-Aldrich, T9424). The densitometry analyses of northern blot sample bands were performed using ImageJ software[78]. The northern blot bands were normalized to the density of SYBR gold-stained control RNA bands (5.8S rRNA, 5S rRNA, Type I and type II tRNA bands) on urea PAGE. To determine if there was an association

between serum cholesterol levels and tsRNA-Glu-CTC expression, a Pearson correlation was performed. Human liver RNAs used for the norther blot analysis were isolated from normal liver tissue of five different donors. Liver RNAs were pooled together and provided by BioChain Institute Inc (R1234149-P).

## PANDORA-seq of small RNAs in mouse liver

PANDORA-seq protocol has been described in detail in our previous reports[22,37]. More recently, we have described a detailed step-by-step protocol of optimized PANDORA-seq and related analysis methods in a recent protocol[35]. Briefly, total RNAs were extracted from mouse liver using TRIzol reagents (Sigma-Aldrich, T9424) as previously described[22,36,37]. The hepatic RNA samples was mixed with an equal volume of 2× RNA loading dye (New England Biolabs, B0363S) and incubated at 75 °C for 5 min. The mixture was loaded into 15% (wt/vol) urea polyacrylamide gel (10 ml mixture containing 7 M urea (Invitrogen, AM9902), 3.75 ml Acrylamide/Bis 19:1, 40% (Ambion, AM9022), 1 ml 10× TBE (Invitrogen, AM9863), 1 g l⁻¹ ammonium persulfate

(Sigma–Aldrich, A3678-25G) and 1 ml l$^{-1}$ TEMED (Thermo Fisher Scientific, BP150-100) and run in a 1× TBE running buffer at 200 V until the bromophenol blue reached ~1 cm from the bottom of the gel. Small RNA of 15–50 nucleotides was visualized with SYBR Gold solution (Invitrogen, S11494) and excised[22,35,37]. A sample of the eluted RNA was stored in −80 °C for Traditional small RNA-seq[22,36,37]. The remaining RNA was eluted and then treated with T4PNK reaction mixture (5 μl 10x PNK buffer, 1 mM ATP, 10U T4PNK) followed by RNA isolation with TRIzol. The collected RNAs were then treated with AlkB mixture (50 mM HEPES, 75 μM ferrous ammonium sulfate, 1 mM α-ketoglutaric acid, 2 mM sodium ascorbate, 50 mg/l BSA, 4 μg/ml AlkB, 2,000 U/ml RNase inhibitor) followed by RNA isolation with TRIzol. The recombinant AlkB enzyme was prepared by Dr. Linlin Zhao (University of California, Riverside, CA) as previously described[22,37]. The adapters (New England Biolabs, E7330S) were ligated sequentially (3′ adapter, revers transcription primer, then 5′ adapter). First-strand cDNA synthesis was performed followed by PCR amplification with PCR Primer Cocktail and PCR Master Mix to enrich the cDNA fragments. Finally, the PCR products were purified from PAGE gel and prepared for sequencing at the Genomics Center of University of California, San Diego (Illumina system)[22,36,37].

Both the traditional small RNA-seq and PANDORA-seq data were processed using the computational framework *SPORTS1.1* with one mismatch tolerance (*SPORTS1.1* parameter setting: -M 1)[27,35]. The *edgeR* tool[96] was applied to identify the differentially expressed small RNA species between the LCD and HCD groups, using the *TMM* method for read counts normalization and the likelihood ratio test for differential expression analysis[22]. The small RNA species with false discovery rate (FDR) < 0.05 were deemed differentially expressed. All the small RNA-seq datasets have been deposited in the Gene Expression Omnibus (GSE300043).

## RNA sequencing and transcriptomic data analysis
Total RNAs were extracted from mouse liver using TRIzol reagents (Sigma-Aldrich, T9424)[22,36,38]. RNA integrity was verified by a Bioanalyzer (Agilent Technologies Inc., Santa Clara, CA). cDNA library construction and sequencing were performed following the Illumina standard operation pipeline and the previously described RNA-seq data analysis methods[36,38,54,97,98]. The *kallisto* tool was applied to quantify protein-coding gene expression from the sequencing data[99] and the *edgeR* tool was applied to identify the differentially expressed genes (DEGs)[96] with the cut-off for differential expression as fold change (FC) > 1.5 and a *P* value < 0.01. Gene Ontology Biological Process (GOBP) analysis was also performed using the definition from GO project. The DAVID bioinformatics tool[100] was applied to detect the GOBP terms enriched by the DEGs. For prioritized GOBP terms, we used the Functional Analysis of Individual Microarray Expression (FAIME) algorithm to calculate geneset scores[101]. A higher FAIME score suggests an increased overall expression of a given GOBP term/geneset. All the RNA-seq datasets have been deposited in the Gene Expression Omnibus (GSE300043).

## Lipid analysis
Serum total cholesterol and triglyceride levels were measured by the Wako Cholesterol E enzymatic colorimetric assay kit (Wako, 999-02601) and the Wako L-type TG M assay kit (Wako, 994-02891) following to the manufacturer's instructions[37,78,102]. The lipoprotein fractions were isolated in a Beckman Colter XPN100-IVD ultracentrifuge as previously described[37,78,102]. Lipid contents were extracted from mouse liver and hepatic cholesterol and triglyceride levels were measured as previously described[77,103,104].

## Atherosclerotic lesion analysis
The atherosclerotic plaque sizes were quantified at the aortic root[37,105,106]. Optimal Cutting Temperature (OCT)-compound-embedded hearts were sectioned at a 12 μm thickness keeping all the three valves of the aortic root in the same plane, and stained with Oil-Red-O. Images were taken and plaque size was quantified using a Nikon microscope Ti2 model (Nikon). For the immunofluorescence staining, samples were fixed in 4% PFA for 15 min and permeabilized with 0.5% Triton X-100 in PBS for 15 min. After incubating with 5% BSA for 1 hr at room temperature, the slides were incubated with primary antibodies against CD68 (Bio-Rad, MCA1957, 1:100) and α-SMA (Abcam, ab5694, 1:100) at 4 °C for 12 to 16 hr[37,105]. The sections were rinsed with PBS and incubated with fluorescein-labeled secondary antibodies (Life Technologies). The nuclei were stained by mounting the slides with DAPI medium (Vector Laboratories). Samples were imaged and analyzed with a Nikon fluorescence Ti2 model microscope.

## RNA isolation and quantitative real-time PCR
Total RNAs were extracted from mouse tissue or cell lines using TRIzol reagents (Sigma-Aldrich, T9424)[37,78]. Total RNAs were reverse transcribed using the Script™ Reverse Transcription Supermix for RT-qPCR (Bio-Rad: 1708841). Quantitative real-time PCR was performed using gene-specific primers (Supplementary Table 3) and the SYBR Green Supermix (Bio-Rad, 172-5124) as previously described[37,78].

## Northern blot
Northern blot was performed as we previously described[22,37]. Briefly, total RNAs were separated by a 10% urea-PAGE gel followed by SYBR gold staining of nucleic acids (Thermo Fischer, S11494) and immediately imaged. The RNAs were then transferred to a positively charged nylon membrane (Roche, 11417240001) and ultraviolet crosslinked with 0.12 J of energy. Membranes were hybridized with PerfectHyb Plus Hybridization Buffer (Sigma-Aldrich, H7033) for 1 hr at 42 °C. To detect tsRNAs and miRNAs, the membranes were incubated with DIG-labeled oligonucleotide probes (Supplementary Table 2) at 42 °C for overnight and the images were taken by using a Bio-Rad Chemidoc Imaging System. The densitometry analyses of northern blot sample bands were performed using ImageJ software and normalized to the density of SYBR gold-stained control RNA bands (5.8S rRNA, 5S rRNA, Type I and type II tRNA bands) on urea PAGE. To calculate probe efficiency factors for miRNA-122 and tsRNA-Glu-CTC (Fig. 1g), the density of northern blot bands normalized with the density of SYBR gold-stained RNA bands on Urea PAGE.

## Western blot
Protein isolation and western blot analysis was performed as previously described[103,107,108]. Briefly, tissue lysate samples were resolved on SDS-PAGE. Proteins were then transferred to nitrocellulose (NC) membrane (Bio-rad, 1620115). The membrane was blocked in phosphate buffered saline solution with 0.05% Tween 20 (PBST, pH 7.4) containing 5% BSA (Sigma, A3294) for 2 hr, and then incubated with primary antibodies against SREBP2 (ProteinTech, 28212-1-AP, 1:1000), PCSK9 (Abcam, ab28870, 1:1000), HMGCR (Novus-bio,NBP2-6688, 1:1000), Insig1 (ProteinTech, 55282-1-AP, 1:1000), Insig 2 (ProteinTech, 24766-1-AP, 1:1000), Scap (ThermoFisher, PA5-28982, 1:1000), SR-BI (Novus-bio, NB400, 1:1000), LDLR (Abcam, ab52818, 1:1000), Actin (Sigma, A2066, 1:2000), or GAPDH (Sigma, G8795, 1:2000) in PBST with 5 % BSA for 2 hr at room temperature. After subsequent three-time washing in PBST, the membranes were developed by Pierce ECL Western Blotting Substrate kit (Thermo Fisher Scientific, 32209), and the images were taken by using a Bio-Rad Chemidoc Imaging System.

## Histological analysis
For hematoxylin and eosin staining, tissues were fixed in 4% neutral buffered formalin and embedded in paraffin. Tissue sections were stained with hematoxylin (Sigma-Aldrich; 1.05175) and eosin (Sigma-Aldrich; R03040) following standard protocols as previously described[102,103]. Oil-Red-O staining of neutral lipids was performed as

previously described[77]. Briefly, liver tissues were embedded in OCT and sectioned at 10 μm thickness. Tissue sections were then dried, fixed in 4% PFA, incubated for 5 min in 60% isopropanol then incubation in 0.3% Oil-Red-O (Sigma-Aldrich; O0625) for 20 min.

## Cell culture and transfection assays

The human hepatic cell line HepG2 was purchased from the American Type Culture Collection (HB-8065)[77,109]. The SREBP2 promoter reporter plasmids (pGL4.23-SREBP2_p) containing 1.4 kb SREBP promoter region (Addgenes, VB180910-1231mwu) and SREBP2 expression plasmids were kindly provided by Dr. Tamer Sallam at University of California, Los Angels[56]. Different primer sets were used to construct reporter plasmids containing 793 bp, 485 bp, 300 bp, and 245 bp of SREBP2 promoter (Supplementary Table 4). The PCR products were subcloned into the same pGL4.23 vectors. The E-box mutant plasmids were generated by using QuikChange II XL Site-Directed Mutagenesis Kit (Aglient, 200521) and primer set (Supplementary Table 4). Reporter transfection assays were performed as described previously[59,78]. Briefly, HepG2 cells were transfected with various luciferase reporter plasmids along with cytomegalovirus X-β-galactosidase control plasmids using FuGENE 6 (Promega Corporation; E2691)[51,59,77,78]. Then the cells were lysed with Passive Lysis Buffer (Promega Corporation; E1941), and extracts were prepared for β-galactosidase and luciferase assays[51,59,77,78]. Luciferase assays were performed as manufacturer's manual (Promega Corporation; E1531) using a Synergy H1 Hybrid Reader (BioTek Instruments, Inc; 11120535)[78]. The lysate was added 100 μL of β-gal solution and incubated at 37 °C for 5-10 min and then stop the reaction with 50 μL of 1 M $Na_2CO_3$. The lysate was then read $OD_{595}$ using a Synergy H1 Hybrid Reader. Reporter gene activity was normalized to the β-gal transfection controls and the results expressed as normalized Relative Light Unit (RLU) per $OD_{595}$ β-gal per minute to facilitate comparisons between plates. Fold activation was calculated relative to solvent controls. The cells were then lysed, and β-gal and luciferase assays were performed as described Fold activation was calculated relative to the controls[59,78]. To overexpress tsRNA-Glu-CTC in HepG2 cells, 100 nM synthetic control oligonucleotides or tsRNA-Glu-CTC oligonucleotides were transfected into cells using Lipofectamine RNAiMAX (Thermo Fisher, 13778100), followed by RNA isolation after 24–36 hr. For ASO transfection assay, HepG2 cells were transfected with 100 nM control ASO or tsRNA-Glu-CTC ASO using Lipofectamine RNAiMAX followed by RNA isolation after 24–36 h.

## RNA fluorescence in situ hybridization (RNA-FISH)

RNA-FISH was performed by using ViewRNA™ ISH Cell Assay Kit (Cat#QVC0001, Invitrogen), according to the manufacturer's instructions. tsRNA-Glu-CTC FISH probe was synthesized and labeled with Type 4 (FITC) by Invitrogen. Briefly, HepG2 cells were fixed with 4% formaldehyde solution for 30 min at room temperature and digested with working protease solution (1:4000) for 10 min at room temperature before they were washed and incubated with probe sets at 40 °C for 3 hr. The samples were then washed and incubated with Pre-Amplifiers solutions at 40 °C for 30 min before they are washed again and incubated with Label probe mix at 40 °C for 30 min. The samples were then stained with DAPI and the images were acquired by using a Zeiss 880 confocal microscope.

## Isolation of tsRNA-Glu-CTC-associated DNA fragments

Isolation of tsRNA-Glu-CTC-associated DNA fragments in HepG2 cells was performed by using Biotin and Streptavidin conjugated Sepharose bead system. Briefly, biotin-labeled tsRNA-Glu-CTC oligonucleotides and control biotin-labeled oligonucleotides were transfected into HepG2 cells. After 24 hr, cells were washed with cold PBS twice and then incubated with 1% PFA at room temperature for 15 mins for the RNA cross-linking. Cells were then incubated with 100 mM Glycine at room temperature for 5 min for stopping PFA induced RNA cross-

linking. Cells were washed with cold PBS twice and then lysed with 500 μL lysis buffer containing 1% SDS, 1 mM EDTA, 50 mM HEPEs (PH7.5), 140 mM NaCl, 1% Triton-X100, 1U/μL RNase inhibitors, and 1X protease inhibitors (Cocktails). Chromatin shearing was used by sonication (Vibra-cellTM, Sonics & Materials Inc.) with 50% AMP for 10 cycles of 30 sec on >30 sec off pulse program. Streptavidin Sepharose beads were washed with 20 mM Tris-Hcl (PH7.5) buffer twice. The chromatin shearing RNA cross-linking solution was incubated with Streptavidin Sepharose beads and rotated on a shaker at room temperature for 1 hr. The samples were then transferred into 1.5 mL Ultrafree-MC tube (Millipore UFC30GV0S) and washed with wash buffer containing 10 mM Tris-Hcl (PH7.5), 1 mM EDTA and 0.1U/ul RNase inhibitors for 4 times. Isolated DNA from the samples were performed by a SimpleChIP Enzymatic Chromatin IP kit (Cell signaling, 9003) as previously described[58,59]. The isolated genomic DNA was used for PCR analysis with the primer sets targeting different promotes including SREBP promoter containing the E-Box motif (Supplementary Table 3).

## Isolation of tsRNA-Glu-CTC-associated proteins

Isolation of tsRNA-Glu-CTC-associated proteins in HepG2 cells was performed by using Biotin labeled tsRNA-Glu-CTC probes (Supplementary Table 2) and Dynabeads M-280 Streptavidin beads system. Briefly, -10×10^6 HepG2 cells were washed with cold PBS twice and were subjected to UV-crosslinking on ice at 80000 uj/cm² for 3 times. The cells were then lysed with 1000 uL tsRNA pulldown lysis buffer containing 1% SDS, 1 mM EDTA, 50 mM HEPEs (PH7.5), 140 mM NaCl, 1% Triton-X100, 1U/μL RNase inhibitors, and 1X protease inhibitors (Cocktails). Completed lysed cells were sonicated (Vibra-cellTM, Sonics & Materials Inc.) with 50% AMP for 3 cycles of 30 sec on a > 30 sec off pulse program. 500 μL RNA-protein lysis solution was then transferred into a RNase-free 2 mL tube followed by adding 20 μL denatured 100 μM biotinylated tsRNA-Glu-CTC probes (Supplementary Table 2), 26 μl 20x SSC solution (Thermo Fisher Scientific, 15557044) and 10 μL RNase inhibitor (New England Biolabs, M0314L), and 444 μL RNase-free $H_2O$. The samples were then incubated on a rotor shaker overnight at 4 °C. 100 ul Dynabeads M-280 Streptavidin beads for each reaction were first washed with 20 mM Tris-Hcl (PH7.5) buffer twice. The tsRNA probe-RNA-proteins solution was then incubated with washed Dynabeads M-280 Streptavidin beads and rotated on a shaker at room temperature for 1 hr. The beads were collected after the DynaMag Magnet separation process and washed with 500ul 0.5X SSC for 6 times. 30 μL lysis buffer and 6 μL 6X SDS loading buffer were added to the samples and the samples were then boiled at 95 °C for 10 mins. The pulldown proteins mixture was used for Western blot analysis.

## Chromatin immunoprecipitation (ChIP)

The SimpleChIP Enzymatic Chromatin IP Kit (Cell Signaling Technology; 9003) was used for ChIP analyses as previously described[58,59]. Anti-SREBP2 antibodies (ab30682, Abcam) were utilized in the ChIP assay. The precipitated genomic DNA was purified using spin column purification kit (Cell Signaling Technology; 14209) followed by PCR analysis with the primers targeting different promoters including SREBP2 promoter containing the E-box motif, MTTP promoter and CYP3A4 promoter (Supplementary Table 3).

## Isolation of endogenous tsRNA-Glu-CTC from mouse liver

Endogenous tsRNA-Glu-CTC was isolated from mouse liver by affinity pulldown assay combined with urea acrylamide/bis gel recovery as previously described[110]. Briefly, total RNAs were isolated from mouse liver using TRIzol reagents (Sigma-Aldrich, T9424) as we previously described[37,78]. Small RNAs ( < 200 nt) were separated by RNA precipitation. 10 mg total RNAs were incubated with 10 ml separation buffer containing 2 ml of 50% PEG 8000, 1 ml of 5 M NaCl on ice for

30 min followed by centrifugation at 15000 g at 4 °C for 15 min. Supernatants were then collected and were incubated with small fragment RNA precipitated buffer with the ratio of 1 part of supernatant: 3 parts of ethanol: 0.1 part of 3 M NaAc solution (InvitrogenTM AM9740): 0.005 part of Linear Acrylamide (Thermo Fisher Scientific, AM9520) at -80 °C overnight. The samples were then centrifuged at 15000 g for 15 mins at 4 °C to recover small RNAs ( < 200 nt). 1 mg small RNA ( < 200 nt) solution was transferred into a RNase-free 2 mL tube followed by adding 15 μL denatured 100 μM biotinylated tsRNA-Glu-CTC DNA probe (Supplementary Table 2), 26 μl 20x SSC solution (Thermo Fisher Scientific, 15557044) and 10 μL RNase inhibitor (New England Biolabs, M0314L). The samples were adjusted to 1 mL with RNase-free H$_2$O and were incubated in a 50 °C water bath overnight. 200 μL Streptavidin Sepharose bead solution (Cytiva 17511301) was transferred to a 1.5 mL Ultrafree-MC tube (Millipore UFC30GV0S), washed with 400 μL 20 mM Tris-HCl (pH7.5) for 3 times, and then centrifuged at 2500 g at room temperature for 30 secs each time. Samples were included with Streptavidin Sepharose beads on the rotation shaker for 1 hr at room temperature, centrifuged at 2500 g for 30 sec, and washed with 500ul 0.5X SSC for 4 to 5 times until the OD$_{260}$ < 0.1 for the collected washing solution. 500 μL RNase-free water was added to the tube and incubated at 65 °C for 20 min followed by centrifuging at 2500 g for 30 sec at room temperature. The RNA elution process was repeated 5 times, and the eluted samples were incubated with small RNAs precipitated solution at -80 °C overnight before centrifugation at 15000 g at 4 °C for 20 min. The precipitated small RNAs were separated on 15% urea acrylamide/bis gel and the correct band of tsRNA-Glu-CTC (30-40 nt) were cut. RNA was then recovered as we previously described[22,37] and confirmed by northern blot.

### Identification of tsRNA-Glu-CTC modifications by Mass spectrometric ladder complementation sequencing

Endogenous tsRNA-Glu-CTC was isolated from mouse liver and RNA modifications were identified by MLC-seq[42]. Isolated endogenous tsRNA-Glu-CTC and it's parental tRNA-Glu-CTC was either untreated or treated with 4 μg/mL AlkB enzyme at 37 °C for 30 min. For AlkB treated RNAs, the RNAs were recovered by Trizol reagents from the enzyme reaction mixture. Untreated or AlkB-treated tsRNA samples were then incubated with 50% (v/v) of formic acid (Fisher Scientific, A117-50) at 40 °C for 1 min, 2 min or 5 min. 170 ng of RNA sample was used for each time point. The reaction mixture was immediately frozen on dry ice followed by lyophilization to dry. The dried samples of three different time points for untreated or AlkB-treated RNA were then combined and suspended in 20 μL nuclease-free, deionized water for LC-MS measurement as previously described[42]. Briefly, each combined acid-hydrolyzed tsRNA sample was individually analyzed on an Orbitrap Exploris 240 mass spectrometer (ThermoFisher Scientific) coupled to a Vanquish Horizon UHPLC using a DNAPac reversed-phase (RP) column (2.1 mm×50 mm, ThermoFisher Scientific) with 2% HFIP and 0.1% DIPEA as eluent A, and methanol, 0.075% HFIP, and 0.0375% DIPEA as eluent B. A gradient of 20% to 80% B over 6.7 min was used for the analysis of intact RNA samples, and from 15% to 35% B over 20 min for acid-degraded samples. The flow rate was 0.2 or 0.4 ml/min, and all separations were performed with the column temperature maintained at 70 °C. RNA samples were analyzed in negative ion full MS mode from 410 m/z to 3200 m/z with a scan rate of 2 spectra/s at 120k resolution at m/z 200. The data were processed using Thermo Bio-Pharma Finder 4.0 (ThermoFisher Scientific), and the detailed data analysis was performed as previously described[42].

### Statistics and reproducibility

All data except the high-throughput sequencing data are presented as the mean ± SEM. Individual pairwise comparisons were analyzed by two-sample, one-tailed or two-tailed Student's t test. One-way analysis of variance (ANOVA) with Bonferroni multiple comparison test was used for analyzing more than two groups. Data analysis was performed using the GraphPad Prism 10 software with statistical significance set at $P < 0.05$. For representative experiments including northern blot analysis and tissue staining images, similar results were obtained in three or more independent experiments.

### Reporting summary

Further information on research design is available in the Nature Portfolio Reporting Summary linked to this article.

## Data availability

The data supporting the findings from this study are available within the manuscript and its supplementary information. All the RNA-seq datasets have been deposited in the Gene Expression Omnibus under the accession code GSE300043. Values for graphs in the figures and Supplementary Figs. are provided in the supplementary Source Data file. Source data are provided with this paper.

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

## Acknowledgements

We thank all lab members for their technical assistance, Dr. Linlin Zhao lab for preparing recombinant AlkB enzyme, and Dr. Tamer Sallam for providing the SREBP2 promoter reporter and expression plasmids. This work was partially supported by National Institutes of Health grants (R35ES035015 and R01HL167206 to C.Z.; R01HG012853 to S.Z.; R01HD092431 and R01ES032024 to Q.C. and T.Z.). R.H. was supported by an American Heart Association predoctoral fellowship (23PRE1018751).

## Author contributions

C.Z. and X.L. conceptualized and designed the research. X.L. performed most of the experiments with the help of R.H., X.Z., S.T., X.Y., J.W., and K.P. X.Y. and J.W. performed experiments to identify RNA modifications under the supervision of S.Z. K.P. and E.C.H. conducted human subject recruitment and sample collection. X.L., R.H., X.Z., X.Y., J.W., H.C.R., E.C.H., S.Z., Q.C., T.Z., and C.Z. contributed to the data analysis. X.L., E.C.H., S.Z., Q.C., T.Z., and C.Z. wrote the manuscript.

## Competing interests

The authors declare the following financial interests which may be considered as potential competing interests: S.Z. is the founder and president of DirectSeq Biosciences, Inc. C.Z. and X.L. are inventors on a provisional patent application filed by the Regents of the University of California. All authors declare no non-financial conflict of interest with regard to this study.
