## [Transparent Peer Review file · Nature Communications]

A cholesterol-responsive hepatic tRNA-derived small RNA regulates cholesterol homeostasis and atherosclerosis development

Corresponding Author: Professor Changcheng Zhou

Version 0:

Reviewer comments:

Reviewer #1

(Remarks to the Author)

The authors have responded adequately to my concerns.

Reviewer #2

(Remarks to the Author)

The vast majority of concerns are not addressed experimentally but nevertheless, on balance of factors, the authors introduce a provocative concept that represents a new way of thinking for the field. The idea that tRNAs can mediate critical metabolic regulatory circuits will be of interest. I also appreciate the tedious effort that went into this work and hope the authors continue to build on their studies to offer more clarifications for the field in the future. A few minor remaining issues:

- 1) Many of the responses cite other data but these studies are not expanded in the discussion. For example, please add clarification in the discussion as to why only the LDL fraction was impacted and how this differs from other studies perturbing SREBPs directly.
- 2) The authors barely mention important work in the past connecting SREBPs, lipids and noncoding RNA. For example, microRNAs (mir 33) and lncRNAs (Lexis).

Reviewer #3

(Remarks to the Author)

All my comments were addressed - I congratulate the authors on the nice work!

We would like to thank editors and reviewers for careful critique of our revised manuscript and the comments. We have responded to each of their comments in the following point-by-point responses. We hope you find the revised manuscript acceptable for publication in *Nature Communications*.

Reviewer #1 (Remarks to the Author):

The authors have responded adequately to my concerns.

Response: We thank the Reviewer for reviewing our revised manuscript.

Reviewer #2 (Remarks to the Author):

The vast majority of concerns are not addressed experimentally but nevertheless, on balance of factors, the authors introduce a provocative concept that represents a new way of thinking for the field. The idea that tRNAs can mediate critical metabolic regulatory circuits will be of interest. I also appreciate the tedious effort that went into this work and hope the authors continue to build on their studies to offer more clarifications for the field in the future. A few minor remaining issues:

Response: We thank the Reviewer for the comments. We are glad that the Reviewer recognized the efforts we put into this work and are also encouraged by the positive comments such as “provocative concept”. We will certainly continue exploring the function and mechanism of these abundant but understudied tsRNAs in cardiovascular disease and other diseases.

1) Many of the responses cite other data but these studies are not expanded in the discussion. For example, please add clarification in the discussion as to why only the LDL fraction was impacted and how this differs from other studies perturbing SREBPs directly.

Response: We thank the Reviewer for the comments. We have revised our manuscript with expanded discussion to cover this topic. The revised manuscript now has a whole paragraph to discuss why only the LDL fraction was impacted in our study and how our study differs from other studies perturbing SREBPs directly.

2) The authors barely mention important work in the past connecting SREBPs, lipids and noncoding RNA. For example, microRNAs (mir 33) and lncRNAs (Lexis).

Response: We thank the Reviewer for the comments. We have cited more papers and included more discussions related to lncRNAs (e.g., LeXis, CHROME) and miRNAs (e.g., miRNA 33, miRNA-148) in the revised manuscript.

Reviewer #3 (Remarks to the Author):

All my comments were addressed - I congratulate the authors on the nice work!

Response: We appreciate the reviewer’s compliment! We also thank the Reviewer for the initial constructive comments which encouraged us to continue this line of work and led to improvement in the manuscript. improve our original manuscript.